# Fam134c and Fam134b shape axonal endoplasmic reticulum architecture in vivo

Francescopaolo Iavarone [1,10 ✉], Marta Zaninello [2,3,4,10], Michela Perrone [5], Mariagrazia Monaco [1], Esther Barth [2,3], Felix Gaedke [3], Maria Teresa Pizzo [1], Giorgia Di Lorenzo [1], Vincenzo Desiderio [6], Eduardo Sommella [7], Fabrizio Merciai [7], Emanuela Salviati [7], Pietro Campiglia [7], Livio Luongo [5], Elvira De Leonibus [1,8 ✉], Elena Rugarli [2,3,4 ✉] & Carmine Settembre [1,9 ✉]

## Abstract

**Endoplasmic reticulum (ER) remodeling is vital for cellular organization. ER-phagy, a selective autophagy targeting ER, plays an important role in maintaining ER morphology and function. The FAM134 protein family, including FAM134A, FAM134B, and FAM134C, mediates ER-phagy. While FAM134B mutations are linked to hereditary sensory and autonomic neuropathy in humans, the physiological role of the other FAM134 proteins remains unknown. To address this, we investigate the roles of FAM134 proteins using single and combined knockouts (KOs) in mice. Single KOs in young mice show no major phenotypes; however, combined *Fam134b* and *Fam134c* deletion (*Fam134b/c^dKO*), but not the combination including *Fam134a* deletion, leads to rapid neuromuscular and somatosensory degeneration, resulting in premature death. *Fam134b/c^dKO* mice show rapid loss of motor and sensory axons in the peripheral nervous system. Long axons from *Fam134b/c^dKO* mice exhibit expanded tubular ER with a transverse ladder-like appearance, whereas no obvious abnormalities are present in cortical ER. Our study unveils the critical roles of FAM134C and FAM134B in the formation of tubular ER network in axons of both motor and sensory neurons.**

**Keywords** FAM134C; FAM134B; Endoplasmic Reticulum; Axon
**Subject Categories** Neuroscience; Organelles

## Introduction

Neurodegenerative diseases result from progressive neuron loss and axonal connections. Axons rely significantly on the endoplasmic reticulum (ER) to maintain proper functionality. The tubular ER network serves several functions in the axon, including lipid biosynthesis, glucose regulation, calcium storage, and facilitates both anterograde and retrograde transport (Ozturk et al, 2020; Yperman and Kuijpers, 2023). Disruptions in ER dynamics are linked to neurodegenerative conditions, especially affecting distal axons (Kuijpers et al, 2023; Ozturk et al, 2020).

ER morphology is regulated by ER-shaping proteins like Reticulons, REEPs, and Atlastins (Hu et al, 2008; Orso et al, 2009; Shibata et al, 2008). Mutations in these proteins are associated with disorders such as hereditary spastic paraplegia and hereditary sensory and autonomic neuropathy (Davidson et al, 2012; Esteves et al, 2014; Fischer et al, 2014; Krols et al, 2019; Kurth et al, 2009; Montenegro et al, 2012; Zuchner et al, 2006).

ER maintenance also involves (macro)autophagy, where ER-phagy receptors aid in ER fragment degradation (Chino and Mizushima, 2020; Mochida and Nakatogawa, 2022; Reggiori and Molinari, 2022).

Among ER-phagy receptors, the FAM134 family (FAM134A, FAM134B, and FAM134C) includes proteins with LC3 Interacting Region domain and Reticulon Homology Domain that modulates ER membrane curvature, leading to the formation of tubules and favoring ER-phagy (Bhaskara et al, 2019; Di Lorenzo et al, 2022; Khaminets et al, 2015; Reggio et al, 2021; Siggel et al, 2021). Despite the well-characterized FAM134B and FAM134C regulatory mechanisms, less is known about FAM134A activity (Berkane et al, 2023; Cinque et al, 2020; Di Lorenzo et al, 2022; Gonzalez et al, 2023).

FAM134B mutations are linked to hereditary sensory and autonomic neuropathy type II (HSAN2), causing childhood-onset sensory reductions, mirrored in Fam134b knockout mice (Chen

[1] Telethon Institute of Genetics and Medicine (TIGEM), Pozzuoli, Italy. [2] Institute for Genetics, Faculty of Mathematics and Natural Sciences, University of Cologne, Cologne, Germany. [3] Cologne Excellence Cluster on Cellular Stress Responses in Aging-Associated Diseases, University of Cologne, Cologne, Germany. [4] Center for Molecular Medicine, University of Cologne, Cologne, Germany. [5] Department of Experimental Medicine, Division of Pharmacology, University of Campania "L. Vanvitelli", Naples, Italy. [6] Department of Experimental Medicine, University of Campania "Luigi Vanvitelli", Via L. Armanni 5, 80138 Naples, Italy. [7] Department of Pharmacy, University of Salerno, Via Giovanni Paolo II, 132, Fisciano, SA 84084, Italy. [8] Institute of Biochemistry and Cell Biology, Monterotondo, Rome, Italy. [9] Department of Clinical Medicine and Surgery, Federico II University, Naples, Italy. [10] These authors contributed equally: Francescopaolo Iavarone, Marta Zaninello. ✉E-mail: f.iavarone@tigem.it; deleonibus@tigem.it; elena.rugarli@uni-koeln.de; settembre@tigem.it

et al, 2023; Khaminets et al, 2015; Kurth et al, 2009) However, the physiological roles of other FAM134 members are less understood.

This study explores FAM134 proteins using mouse models with single and combined genetic deletions. While individual FAM134 protein deletions are tolerated in young mice, simultaneous deletion of FAM134B and FAM134C results in severe motor and sensory degeneration due to improper ER orientation in axons.

## Results and discussion

### Concomitant deletion of *Fam134b* and *Fam134c*, but not of *Fam134a*, leads to growth defects and early-onset neurodegeneration

To investigate the physiological role and functional redundancy of the FAM134 proteins (FAM134A, B, and C), we generated knockout mice for each individual member (*Fam134a^{KO}*, *Fam134b^{KO}*, and *Fam134c^{KO}*) and then crossed these to obtain double knockout (^{dKO}) combinations (*Fam134a/b^{dKO}*, *Fam134a/c^{dKO}*, *Fam134b/c^{dKO}*). Reduced body weight was observed in *Fam134b/c^{dKO}* mice as compared with all the other genotypes both at 4 and 15 weeks of age (Figs. 1A and EV1A). Skeletal and cardiac muscle weight were reduced in *Fam134b/c^{dKO}* compared with single KOs and WT mice (Figs. 1B–D and EV1B,C). Furthermore, skeletal elements such as the femur, tibia, humerus, and scapula showed a decreased length in 4-week-old *Fam134b/c^{dKO}* mice compared with controls (Fig. 1E). Collectively, these data suggested a growth defect in *Fam134b/c^{dKO}* mice. Male and female *Fam134b/c^{dKO}* mice showed behavioral signs of progressive neurodegeneration characterized by evident posterior paralysis, aching posture, and hindlimb clasping (Fig. 1F–H). In contrast, mice of the other genotypes showed no signs of neurological impairment (Fig. EV1D). Remarkably, the lifespan of *Fam134b/c^{dKO}* mice was never longer than 25 weeks (Fig. 1I).

We conducted a battery of behavioral tests to measure both motor and sensory functions in *Fam134b^{KO}*, *Fam134c^{KO}*, and *Fam134b/c^{dKO}* mice, comparing them with age- and sex-matched control mice. Exploratory behavior was not affected in young (4-week-old) mutant mice relative to the control animals, as assessed by evaluating the distance traveled and the maximal movement speed in open field tests (Fig. EV1E–G; Table 1). However, *Fam134b/c^{dKO}* mice, but not single KO animals, showed a drop in both traveled distance and maximum speed at 15 weeks of age, indicating an age-dependent decline in motor ability (Fig. EV1E–G; Table 1). The behavioral performance in tests requiring neuromuscular strength, such as hanging wire and hanging steel tests was already affected at 4 weeks of age (Fig. 1J,K; Table 1). Similarly, in tasks requiring motor endurance capacity and equilibrium, such as running on an accelerating rod, *Fam134b/c^{dKO}* mice were already impaired at 4 weeks (Fig. 1L; Table 1).

Nociception was evaluated using the hot plate test. *Fam134b^{KO}* and *Fam134c^{KO}* did not show any significant difference compared with WT mice in response to the heat stimulus at both 4 and 15 weeks of age, while *Fam134b/c^{dKO}* mice showed an early-onset of delayed withdrawal response, suggesting hypoalgesia (Fig. EV1H; Table 1).

These data indicate that the concomitant deletion of *Fam134b* nor *Fam134c* dramatically impacts motor and sensory functions in mice. Although the importance of FAM134B in assuring a proper somatosensory transmission was expected, it was instead surprising to find a significant involvement of motor circuits in *Fam134b/c^{dKO}* mice.

We next analyzed somatosensory perception by measuring the firing of spinal nociceptive specific neurons in *Fam134b/c^{dKO}*, *Fam134b^{KO}*, *Fam134c^{KO}*, and WT mice aged 4 weeks. The firing of these neurons is expressed as firing rate, frequency, and duration of excitation. In the WT group, the neuronal cell population had a firing rate of $0.40 \pm 0.053$ spikes/s, a burst frequency of $4.572 \pm 0.313$ Hz and a duration of excitation of $2.548 \pm 0.049$ s (Fig. EV1I–K; Appendix Fig. S1). No significant changes in firing rate ($0.462 \pm 0.062$ spikes/s, $P > 0.9999$), burst frequency ($6.858 \pm 0.951$ Hz, $P = 0.98$), and burst duration ($2.546 \pm 0.053$ s, $P > 0.9999$) were observed in *Fam134b^{KO}* mice compared with the WT group (Fig. EV1I–K; Appendix Fig. S1). *Fam134c^{KO}* mice showed a slight but significant increase in firing rate ($4.107 \pm 0.599$ spikes/s, $P = 0.0455$), burst frequency ($34.586 \pm 5.385$ Hz, $P < 0.0001$), and burst duration ($3.184 \pm 0.094$ s, $P < 0.0001$) compared with the WT group (Fig. EV1I–K; Appendix Fig. S1). *Fam134b/c^{dKO}* mice showed a remarkable increase in firing rate ($13.565 \pm 2.118$ spikes/s, $P < 0.0001$; $F_{3,189} = 52.70$, $P < 0.0001$), in frequency of excitation ($54.984 \pm 7.761$ Hz, $P < 0.0001$; $F_{3,189} = 31.58$, $P < 0.0001$), and in the duration of excitation ($3.239 \pm 0.129$ s, $P < 0.0001$; $F_{3,189} = 3.923$, $P < 0.0001$), compared with the WT group (Fig. EV1I–K; Appendix Fig. S1). These data are consistent with *Fam134b/c^{dKO}* mice experiencing degeneration of tactile fibers, determining a pain-related synaptic alteration.

### *Fam134b/c^{dKO}* mice undergo axonal degeneration and NMJ alterations

The observed motor and sensory deficits in *Fam134b/c^{dKO}* mice indicated a severe neurological impairment. We did not observe any major alterations in the tissue architecture analyzing brain sections from young WT and *Fam134b/c^{dKO}* mice (Appendix Fig. S2). Immunofluorescence analysis using glia, microglia, and neuron markers revealed no evidence of neuroinflammation and/or altered neuronal density (Appendix Fig. S3A,B). Furthermore, the dendrite marker MAP2 also appeared similarly distributed in the granular cell layer of WT and *Fam134b/c^{dKO}* cerebellum (Appendix Fig. S3C). We also analyzed motoneurons in the lumbar spinal cord and sensory neurons in the lumbar dorsal root ganglia (DRG) of 4-week-old WT and *Fam134b/c^{dKO}* mice, and we found no notable differences relative to these neuronal populations in *Fam134b/c^{dKO}* mice at 4 weeks of age (Appendix Fig. S4A,B). Similarly, when we examined the density of different types of sensory neurons in the DRG, we found that *Fam134b/c^{dKO}* neurons exhibited a soma density comparable with WT mice, although ganglia appeared slightly smaller than controls (Appendix Fig. S4C,D).

Next, we examined the axonal integrity of the sciatic nerve, which serves as a major conduit for sensory and motor signals between the central nervous system and the periphery. At 4 weeks, the number and size of axons in *Fam134b^{KO}* and *Fam134c^{KO}* nerves were comparable with WT, while the *Fam134b/c^{dKO}* sciatic nerves exhibited initial signs of axonal degeneration, and displayed an increased percentage of smaller axons at the expense of larger ones (Fig. 2A–D). By 15 weeks, most of the axons in the *Fam134b/c^{dKO}* nerve had undergone massive degeneration, resulting in a substantial loss in their number (Fig. 2E–G), which mainly affected

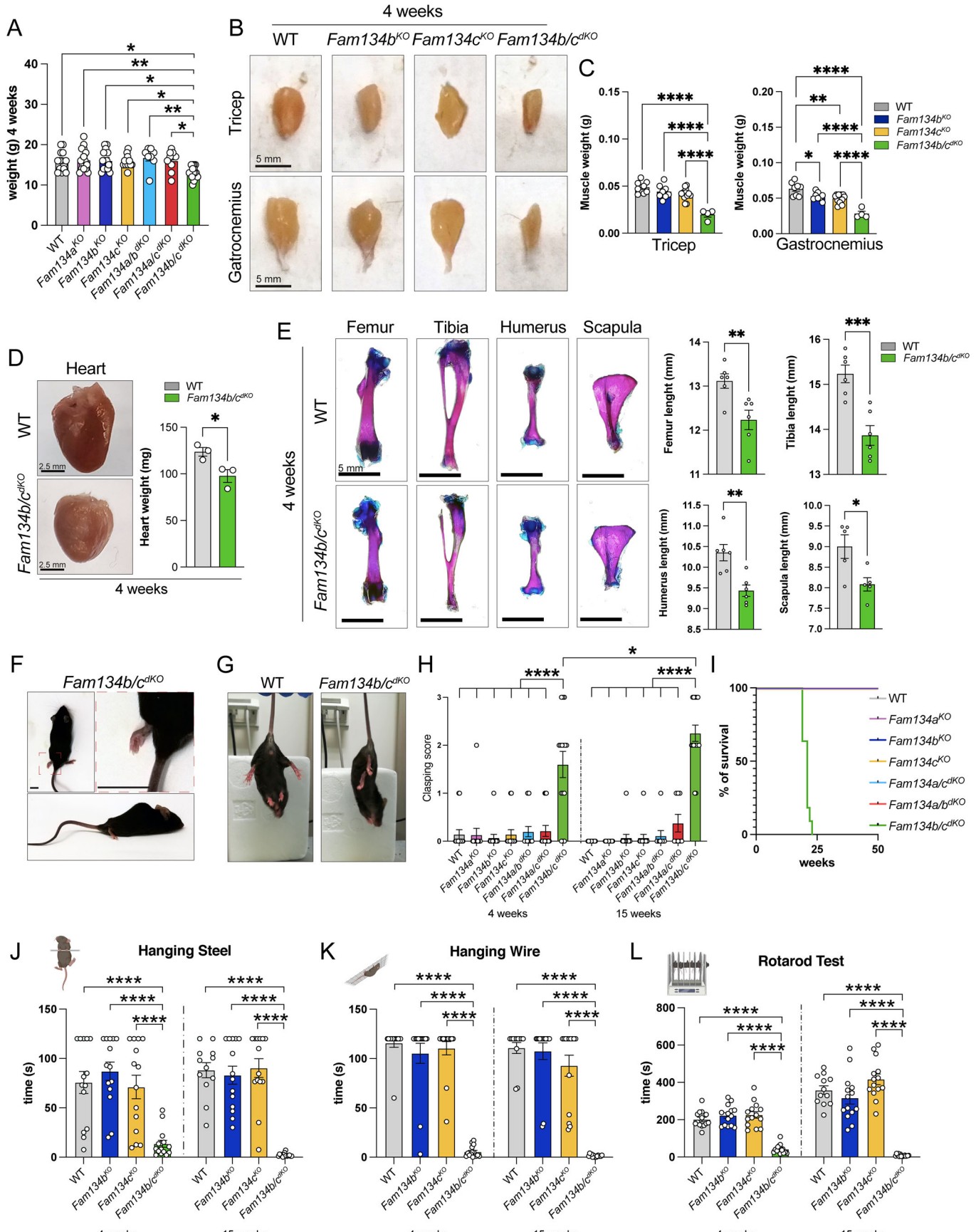

◄   **Figure 1. *Fam134b/c* double KO mice exhibit defective growth and neurological impairment.**

(A) *Fam134b/c* double KO (*Fam134b/c^dKO^*) mice show reduced body weight at 4 weeks of age, as compared to the other Fam134 mutant mice. $n \geq 8$ animals/group. Statistical significance was determined by one-way ANOVA followed by Tukey's multiple comparisons test. Data represent mean ± SEM. *$P < 0.05$, **$P < 0.01$. (B) Representative images of tricep and gastrocnemius muscles from WT, *Fam134b^KO^*, *Fam134c^KO^* and *Fam134b/c^dKO^* mice aged 4 weeks. Scale bar, 5 mm. (C) The weight of tricep and gastrocnemius muscles are massively reduced in *Fam134b/c^dKO^* mice. $n \geq 4$ animals/group. Statistical significance was determined by one-way ANOVA followed by Tukey's multiple comparisons test. Data represent mean ± SEM. *$P < 0.05$, **$P < 0.01$, ****$P < 0.0001$. (D) Representative images of cardiac muscle from WT and *Fam134b/c^dKO^* mice aged 4 weeks, showing a reduced weight in *Fam134b/c^dKO^* compared with WT. $n = 3$ animals/group. Statistical significance was determined by Student's *t*-test. Data represent mean ± SEM. *$P = 0.0348$. (E) Representative images of the femur, tibia, humerus, and scapula from WT and *Fam134b/c^dKO^* mice aged 4 weeks, showing decreased length in *Fam134b/c^dKO^* compared with WT mice. Scale bar, 5 mm. $n \geq 5$ animals/group. Statistical significance was determined by Student's *t*-test. Data represent mean ± SEM. *$P = 0.0235$, **$P = 0.0034$ (humerus), **$P = 0.0094$ (femur), ***$P = 0.0009$. (F) Representative images of aching posture and loss of hindlimb mobility in *Fam134b/c^dKO^* mice. Scale bars, 1 cm. (G) Representative images of 4-week-old *Fam134b/c^dKO^* and WT mice showing hindlimb clasping in *Fam134b/c^dKO^* mice while they are suspended by the tail. (H) Clasping test scores in 4-week-old and 15-week-old mice with the indicated genotypes displaying an early-onset and progressive increase of the clasping sign in *Fam134b/c^dKO^* mice compared with the other indicated genotypes. The score assigned ranges from 0 (no alterations) to 3 (severe alterations). $n \geq 12$ animals/group. Statistical significance was determined by one-way ANOVA, followed by Tukey's multiple comparisons test. Data represent mean ± SEM. *$P < 0.05$, ****$P < 0.0001$. (I) Kaplan–Meier survival curves relative to all the indicated mouse models showing a reduced lifespan of *Fam134b/c^dKO^* mice. $n \geq 5$ animals/group. (J–L) Evaluation of motor activity showing *Fam134b/c^dKO^* mice fail to complete hanging steel (J), hanging wire (K), and rotarod (L) test at both time points analyzed, compared with *Fam134b^KO^*, *Fam134c^KO^*, and WT mice. $n \geq 12$ animals/group. Statistical significance was determined by one-way ANOVA, followed by Tukey's multiple comparisons test. ****$P < 0.0001$. Data information: Statistical analysis and exact *P* values are included in the source data files. Images of behavioral tests were generated using Biorender. Source data are available online for this figure.

axons of larger size (Fig. 2H). Notably, *Fam134b/c^dKO^* mice accumulated degenerating axons in a time-dependent manner also in the dorsal funiculi of the spinal cord, which contain long ascending proprioceptive axons (Appendix Fig. S5A–C). Electron microscopy analysis in the sciatic nerve at 15 weeks revealed several large myelinated fibers containing a dark cytoplasm with accumulation of organelles and cytoskeletal components, a sign that precedes neurodegeneration (Fig. 2I; Appendix Fig. S6A). Consistently, large areas of the sciatic nerve were occupied by collagen fibers instead of axons (Fig. 2I; Appendix Fig. S6A). In addition, axonal pathology was also observed at the level of unmyelinated fibers in the Remak bundles, where axons contained abnormal material (Fig. 2J; Appendix Fig. S6B).

Consistent with these signs of degeneration, analysis of *Fam134b/c^dKO^* sciatic nerve cross-sections revealed a reduced area compared with single KO and WT mice at both 4 (Fig. EV2A) and 15 weeks (Fig. 3A). To unravel which types of axon degenerate, we labeled sciatic nerve sections with NF200 (Neurofilament 200 kDa), a marker of sensory myelinated axons, and ChAT (choline acetyltransferase), an enzyme predominantly synthesized in motor axons (Engel et al, 1980) (Figs. 3B and EV2B). At 4 weeks, the number of axons identified as NF200⁺ or ChAT⁺ did not differ significantly between *Fam134b/c^dKO^* and WT mice (Fig. EV2C,D), whereas a notable reduction in the number of both types of axons was evident in the *Fam134b/c^dKO^* nerve at 15 weeks (Fig. 3C,D). Particularly noteworthy was the distribution of axon diameters in both NF200⁺ and ChAT⁺ axons, indicating an accumulation of smaller axons in *Fam134b/c^dKO^* mice as early as 4 weeks (Fig. EV2E,F), in agreement with the semithin sections. A similar pattern emerged at 15 weeks, but with fewer axons remaining in the nerve, suggesting progressive degeneration (Fig. 3E,F). These data indicate that both sensory and motor fibers are affected in *Fam134b/c^dKO^* mice.

Next, we analyzed neuromuscular junctions (NMJ), which are the critical communication points between motoneurons and skeletal muscle fibers. We performed immunofluorescence analysis of NF200, AChR, and Synaptophysin in the Extensor Digitorum Longus (EDL) muscle aged 4 or 15 weeks (Figs. 3G and EV2G; Movies EV1–EV8) in order to measure: (i) NMJ innervation

(NF200⁺), (ii) morphometric parameters of the NMJs (AChR⁺ clusters), (iii) the nerve terminals (Synaptophysin 1⁺ clusters) in the Extensor Digitorum Longus (EDL) muscle at both 4 and 15 weeks (Figs. 3H–K and EV2H–K). At 4 weeks, there was a notable increase in the percentage of NMJ denervation in *Fam134b/c^dKO^* mice compared with single KO and WT mice (Fig. EV2H) that became even more pronounced at 15 weeks (Fig. 3H), consistent with the progressive axonal degeneration observed in the sciatic nerve (Fig. 3B–F). While area quantification of AChR⁺ clusters showed no significant differences between any of the mice at both time points (Figs. 3I and EV2I), the morphology of *Fam134b/c^dKO^* NMJs appeared visibly altered. This was indicated by fewer branching points at both time points analyzed (Figs. 3J and EV2J), suggesting that *Fam134b/c^dKO^* NMJs had lost their proper structural complexity. Furthermore, at both time points, the ratio between the presynaptic (Synaptophysin 1⁺ cluster) and postsynaptic (AChR⁺ cluster) area was significantly decreased in *Fam134b/c^dKO^* mice compared to the other groups (Figs. 3K and EV2K). These results were consistent with the increased NMJ denervation observed in *Fam134b/c^dKO^* muscle (Figs. 3H and EV2H). The dysfunctional NMJs observed in *Fam134b/c^dKO^* mice can lead to impaired neuromuscular signaling, in turn promoting muscle atrophy. Collectively, these findings can account for the severe motor impairment phenotype in *Fam134b/c^dKO^* mice.

## FAM134B and FAM134C control tubular ER organization within axons

Given the role of FAM134 proteins in ER-phagy, we examined the ultrastructural morphology of the ER in large, myelinated axons of the sciatic nerve. As expected, the ER appears as dispersed short tubules in most WT axons cut transversally, reflecting its longitudinal distribution (classified as normal; Appendix Fig. S7A) at both 4 and 15 weeks (Fig. 4A–C; Appendix Fig. S7B). In contrast, we observed an increase in the percentage of axons containing longer ER tubules, which in some cases were running transversely to the axonal length (classified as intermediate or accumulated, depending on the extent of the phenotype; Appendix Fig. S7A) in mice carrying a deletion of FAM134 proteins

**Table 1. Detailed information for the analysis of behavioral measures.**

| Measure | Statistical test | Sample size |
|---|---|---|
| Body weight<br>Figure 1A<br>Figure EV1A | One-way ANOVA (Factor: genotype, 7 levels: WT, $Fam134a^{KO}$, $Fam134b^{KO}$, $Fam134c^{KO}$, $Fam134a/b^{dKO}$, $Fam134a/c^{dKO}$, $Fam134b/c^{dKO}$)<br>4 weeks: $F_{1,78} = 5.850$; $P < 0.0001$;<br>15 weeks: $F_{1,78} = 13.573$; $P < 0.0001$. | 4 weeks:<br>WT = 13 $Fam134a^{KO}$ = 11 $Fam134b^{KO}$ = 14<br>$Fam134c^{KO}$ = 14 $Fam134a/b^{dKO}$ = 9 $Fam134a/c^{dKO}$ = 9<br>$Fam134b/c^{dKO}$ = 15<br>15 weeks:<br>WT = 13 $Fam134a^{KO}$ = 11 $Fam134b^{KO}$ = 14<br>$Fam134c^{KO}$ = 14 $Fam134a/b^{dKO}$ = 9 $Fam134a/c^{dKO}$ = 9<br>$Fam134b/c^{dKO}$ = 15 |
| Clasping score<br>Figure 1H | Two-way ANOVA (factor: genotype, 4 levels: WT, $Fam134b^{KO}$, $Fam134c^{KO}$, $Fam134b/c^{dKO}$; factor: age, 2 levels: 4 weeks, 15 weeks). Genotype $F_{1,76} = 45.329$, $P < 0.0001$; age n.s.; genotype × age n.s. | WT = 12 $Fam134a^{KO}$ = 11 $Fam134b^{KO}$ = 14<br>$Fam134c^{KO}$ = 14 $Fam134a/b^{dKO}$ = 9 $Fam134a/c^{dKO}$ = 8<br>$Fam134b/c^{dKO}$ = 15 |
| Total distance traveled<br>(open field test)<br>Figure EV1E | Two-way ANOVA (factor: genotype, 4 levels: WT, $Fam134b^{KO}$, $Fam134c^{KO}$, $Fam134b/c^{dKO}$; factor: age, 2 levels: 4 weeks, 15 weeks). Genotype $F_{1,51} = 11.626$, $P = <0.0001$; age $F_{1,51} = 8.285$, $P = 0.005$; $F_{1,51} = 10.7706$, $P = <0.0001$; genotype × age $F_{1,51} = 10.770$, $P < 0.0001$. | WT = 12; $Fam134b^{KO}$ = 14; $Fam134c^{KO}$ = 14;<br>$Fam134b/c^{dKO}$ = 15. |
| Maximum speed<br>(open field test<br>Figure EV1F | Two-way ANOVA (factor: genotype, 4 levels: WT, $Fam134b^{KO}$, $Fam134c^{KO}$, $Fam134b/c^{dKO}$; factor: age, 2 levels: 4 weeks, 15 weeks). Genotype $F_{1,51} = 12.497$, $P = <0.0001$; age $F_{1,51} = 6.664$, $P = 0.0128$, $P = 0.005$; $F_{1,51} = 3.400$, $P = 0.024$; genotype × age $F_{1,51} = 3.400$, $P = 0.024$. | |
| Latency<br>(Hanging steel test)<br>Figure 1J | Two-way ANOVA (Factor: genotype, 4 levels: WT, $Fam134b^{KO}$, $Fam134c^{KO}$, $Fam134b/c^{dKO}$; factor: age, 2 levels: 4 weeks, 15 weeks). Genotype $F_{1,51} = 40.085$, $P < 0.0001$; age $F_{1,51} = 10.396$, $P < 0.0001$; genotype × age n.s. | |
| Latency<br>(hanging wire test)<br>Figure 1K | Two-way ANOVA (factor: genotype, 4 levels: WT, $Fam134b^{KO}$, $Fam134c^{KO}$, $Fam134b/c^{dKO}$; factor: age, 2 levels: 4 weeks, 15 weeks). Genotype $F_{1,51} = 105.495$, $P < 0.0001$; age n.s.; genotype × age n.s. | |
| Latency (Rotarod test)<br>Figure 1L | Two-way ANOVA (Factor: genotype, 4 levels: WT, $Fam134b^{KO}$, $Fam134c^{KO}$, $Fam134b/c^{dKO}$; factor: age, 2 levels: 4 weeks, 15 weeks). Genotype $F_{1,51} = 76.917$, $P < 0.0001$; age $F_{1,51} = 65.839$, $P < 0.0001$; genotype × age $F_{1,51} = 16.558$, $P < 0.0001$. | |
| Latency (hot plate test)<br>Figure EV1H | Two-way ANOVA (Factor: genotype, 4 levels: WT, $Fam134b^{KO}$, $Fam134c^{KO}$, $Fam134b/c^{dKO}$; factor: age, 2 levels: 4 weeks, 15 weeks). Genotype $F_{1,51} = 21.699$, $P < 0.0001$; age $F_{1,51} = 35.244$, $P < 0.0001$; genotype × age n.s. | |

(Fig. 4A–C; Appendix Fig. S7B). $Fam134b^{KO}$ and $Fam134c^{KO}$ mice showed an increase in axons with an intermediate phenotype, which remained constant between 4 and 15 weeks (Fig. 4A–C; Appendix Fig. S7B). Notably, $Fam134b/c^{dKO}$ mice displayed a pronounced phenotype at 4 weeks, with ~75% of axons belonging to the intermediate and accumulated classes (Fig. 4A,B). In more affected axons, the neurofilament and microtubule distribution also appeared altered and disorganized (Fig. 4A; Appendix Fig. S7B). At 15 weeks, the extreme degeneration observed in $Fam134b/c^{dKO}$ mice precluded further analysis (Fig. 2I; Appendix Fig. S6A). Longitudinal axons of 4-week-old $Fam134b/c^{dKO}$ sciatic nerve were characterized by the presence of ER-ladder-like structures that were absent in WT samples (Fig. 4D). 3-D reconstruction showed that ER tubules are parallel to the axonal length in WT sciatic nerve (Fig. 4E; Movies EV9 and EV10), whereas they are distributed transversely with respect to the longitudinal axon in $Fam134b/c^{dKO}$ sciatic nerve (Fig. 4E; Movies EV11 and EV12). These findings were also observed in longitudinal axons of the spinal cord's ventral horn (Appendix Fig. S7C).

The thickness of myelinated axons measured using g-ratios was unaffected in mice of all genotypes, suggesting that Schwann cells could still myelinate (Fig. EV3A). At 15 weeks, the advanced degeneration observed in $Fam134b/c^{dKO}$ nerve precluded the analysis, while a similar g-ratio was observed in WT, $Fam134b^{KO}$, and

$Fam134c^{KO}$ axons (Fig. EV3B). Moreover, we did not observe obvious alterations in the ER ultrastructure of $Fam134b/c^{dKO}$ Schwann cells (Fig. EV3C). These findings suggest a cell-autonomous dysfunction as a main mechanism driving axonal degeneration in $Fam134b/c^{dKO}$ sciatic nerve.

EM analysis of both lower motoneurons and sensory neurons in DRG showed no obvious alterations of the cortical ER, further indicating that the primary role of the $Fam134b/c$ proteins is in the regulation of axonal ER organization, rather than within the neuronal cell bodies (Fig. EV3D,E).

## $Fam134b/c^{dKO}$ sciatic nerves exhibit structural and metabolic dysfunctions

To characterize the molecular alterations accounting for the axonal phenotype in $Fam134b/c^{dKO}$ mice, we isolated brain, lumbar spinal cord, lumbar DRG and sciatic nerves from 2-week-old $Fam134a^{KO}$, $Fam134b^{KO}$, $Fam134c^{KO}$, $Fam134a/b^{dKO}$, $Fam134a/c^{dKO}$, and $Fam134b/c^{dKO}$ mice. Western blotting analysis of FAM134 proteins demonstrated that while all three proteins were present in the brain, spinal cord, and DRG, only Fam134b and Fam134c were clearly detected in wild-type sciatic nerves, indicating that Fam134a was either not expressed or expressed at very low levels in this tissue (Fig. 5A). Interestingly, Fam134a protein was upregulated in

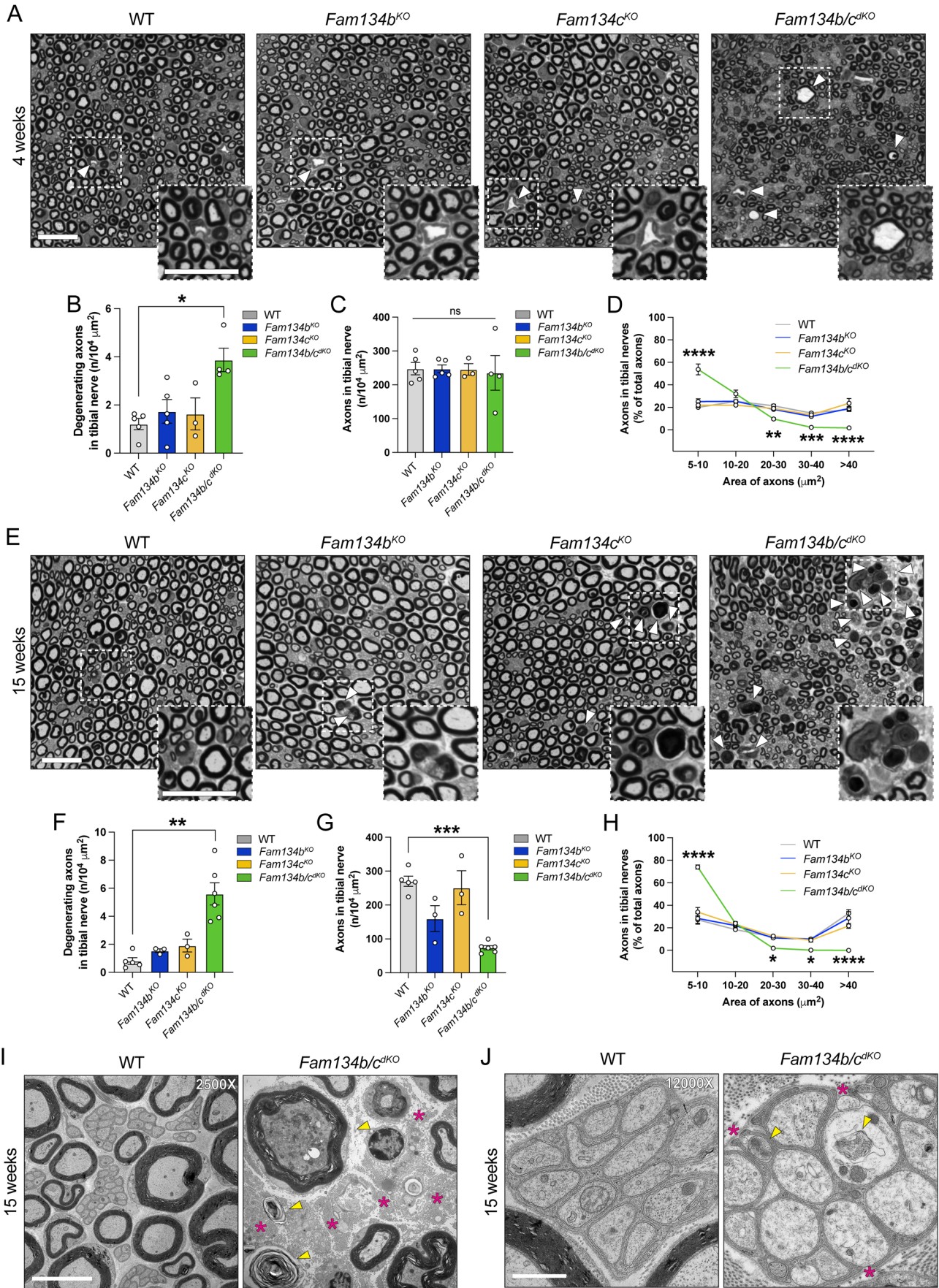

◄ **Figure 2. *Fam134b/c^dKO* sciatic nerves show progressive degeneration.**

(A) Representative toluidine staining of tibial nerves of WT, *Fam134b^KO*, *Fam134c^KO*, and *Fam134b/c^dKO* mice aged 4 weeks. White arrowheads indicate degenerating axons. Scale bar, 20 μm. (B–D) *Fam134b/c^dKO* mice show a mild degeneration of axons in tibial nerves (B, C) and an increased number of axons with small diameter (D) respect with WT, *Fam134b^KO*, and *Fam134c^KO* mice aged 4 weeks. (E) Representative toluidine staining of tibial nerves of WT, *Fam134b^KO*, *Fam134c^KO*, and *Fam134b/c^dKO* mice aged 15 weeks. White arrowheads indicate degenerating axons. Scale bar, 20 μm. (F–H) *Fam134b/c^dKO* mice show increased degeneration of axons in tibial nerves (F), a reduction of axons (G), and an increased number of axons with small diameter (H) compared with WT, *Fam134b^KO*, and *Fam134c^KO* mice. (I) Representative electron micrographs showing degenerating axons accumulating organelles and cytoskeletal components (yellow arrows) in sciatic nerves of WT and *Fam134b/c^dKO* mice aged 15 weeks. Collagen areas are indicated by magenta asterisks. Scale bar, 5 μm. (J) Unmyelinated axons exhibit accumulation of material (yellow arrows) and incomplete wrapping by Schwann cells (magenta asterisks) in tibial nerves of *Fam134b/c^dKO* aged 15 weeks. Scale bar, 1 μm. Data information: $n \geq 3$ animals/group in all experiments. Statistical significance was determined by one-way ANOVA (B, C, F, G) followed by Dunn's multiple comparison test (B) or Dunnett's T3 multiple comparisons test (C, F, G); or two-way ANOVA (D, H) followed by Sidak's multiple comparisons test. Data represent mean ± SEM. ns $P > 0.05$, *$P < 0.05$, **$P < 0.01$, ***$P < 0.001$, ****$P < 0.0001$. Statistical analysis and exact $P$ values are included in the source data files. Source data are available online for this figure.

sciatic nerves isolated from *Fam134b^dKO*, *Fam134c^dKO*, and *Fam134b/c^dKO* mice, potentially as an attempt to compensate for the lack of Fam134b and Fam134c or as a result of impaired ER-phagy mediated degradation (Fig. 5A). These observations provide a plausible explanation for the sciatic nerves being the most affected tissue in the nervous system in *Fam134b/c^dKO* mice. Markers of general autophagy (Sqstm1/p62 and Lc3bII) were not significantly altered in *Fam134b^dKO*, *Fam134c^dKO*, and *Fam134b/c^dKO*, compared with WT (Fig. EV4A,B). Western blot analysis of ER membrane protein Calnexin, a FAM134B substrate (Forrester et al, 2019; Fregno et al, 2018), showed a significant increase in *Fam134b/c^dKO* compared with *Fam134b^dKO*, *Fam134c^dKO*, and WT sciatic nerves (Fig. EV4A,C). Levels of ER-phagy receptors Atl3, Tex264, and Ccpg1 were unaltered in sciatic nerves of *Fam134b/c^dKO*, *Fam134b^dKO*, and *Fam134c^dKO* compared with WT (Fig. EV4A,D), suggesting that their turnover is not obviously dependent on FAM134-mediated ER-phagy. Interestingly, the ER-phagy receptor Rtn3 was significantly downregulated in *Fam134b/c^dKO* compared with *Fam134b^dKO*, *Fam134c^dKO*, and WT sciatic nerves (Fig. EV4A,D).

These data prompted us to study in a more comprehensive manner the alterations occurring in *Fam134b/c^dKO* sciatic nerve. To eliminate potential bias arising from altered axon density and pinpoint the early events contributing to axonal degeneration in the *Fam134b/c^dKO* phenotype, we performed proteomic and lipidomic analysis of sciatic nerve in 2-week-old *Fam134b/c^dKO*, *Fam134b^KO*, *Fam134c^KO*, and WT mice. We selected this age since we observed that the nerve area, axon density, and size in *Fam134b/c^dKO* mice were not yet altered compared with the WT nerve (Fig. EV4E–J).

Principal component analysis (PCA) and 2-dimension hierarchical clustering of all significantly deregulated proteins highlighted distinct proteome patterns, with the *Fam134b/c^dKO* group clearly segregating from the other groups (Appendix Fig. S8A,B). Compared with the WT group, we identified 237, 163, and 435 significantly differentially expressed proteins (DEPs) in *Fam134b^KO*, *Fam134c^KO*, and *Fam134b/c^dKO*, respectively (Fig. 5B). A small number of DEPs were in common in the different comparisons (Fig. 5B). Among the 435 DEPs in the *Fam134b/c^dKO* group, 271 were downregulated, and 164 were upregulated (Dataset EV1). Gene Ontology Cellular Component (GO-CC) enrichment of *Fam134b/c^dKO* DEPs revealed altered protein expression related to the cytoskeleton (22% of all DEPs), mitochondrion (18%), ER (13%), synapse (12%), axon (7%), and myelin sheath (5%) (Fig. 5C). The ER cluster included 55 significant DEPs, with 28 being downregulated and 27 upregulated, almost uniquely in *Fam134b/c^dKO* (Fig. 5D), consistent with a potential functional compensation between Fam134b and Fam134c, at least at this time point.

Gene Ontology Biological Process (GO-BP) enrichment analysis on *Fam134b/c^dKO* ER-located DEPs highlighted overrepresented proteins related to ER tubular network organization (Fig. 5E), which includes known ER-shaping proteins like Reep1, Reep2, Lnpk, and Rab10. We observed a significant downregulation of Reep1 and Reep2 in *Fam134b/c^dKO*, which was validated also by western blot analysis (Fig. 5F,G). Reep1 and Reep2 are an important determinant of ER tubular structure and have been associated with HSP as well as distal hereditary motor neuropathy (Beetz et al, 2013; Beetz et al, 2012; Esteves et al, 2014; Park et al, 2010). Finally, Lnpk stabilizes ER curvature within tubular three-way junctions (Chen et al, 2015), and its mutations have been associated with severe neurodevelopmental disease with epilepsy (Breuss et al, 2018). The analysis also pointed to altered calcium ion transport from the ER lumen to the cytosol (Fig. 5E) and highlighted the involvement of various upregulated proteins (e.g., Ryr1, Casq1, Atp2a1, and Atp2a3; Fig. 5D), providing insights into excitatory transmission. Remarkably, Ryr1 has been recently described as upregulated in Atg5 KO murine neurons, leading to neuron hyperexcitability with consequent neurotoxicity (Kuijpers et al, 2021).

A broader GO-BP analysis displayed enrichments related to "organization" encompassed organelles (i.e., ER and mitochondria) and the cytoskeleton (i.e., "neurofilament bundle assembly"), as well as proteins involved in "paranodal junction assembly", suggesting a massive axonal structure alteration (Appendix Fig. S8C).

Neurofilament heavy chain (NF200) phosphorylation ensures the proper neurofilament assembly in healthy neurons (Laser-Azogui et al, 2015). We found that NF200 in *Fam134b/c^dKO* sciatic nerve was significantly downregulated and hypo-phosphorylated in the *Fam134b/c^dKO* compared with *Fam134b^KO*, *Fam134c^KO*, and WT sciatic nerve (Fig. 5H,I). These results indicated the presence of dysfunctional NF200 in *Fam134b/c^dKO* axons. Other cytoskeletal proteins such as beta-III-tubulin and beta-actin revealed no changes in any genotypes analyzed (Fig. 5H,I).

In addition, Gene Ontology Molecular Function (GO-MF) analysis revealed a deregulation of "microfilament motor activity", which included various myosin isoforms (i.e., Myo1d, Myh1, Myh4, Myh6, Myh8, and Myh13; Appendix Fig. S8D). Altogether, these findings align with the electron microscopy results, which displayed cytoskeletal alterations in some *Fam134b/c^dKO* axons (Fig. 4A).

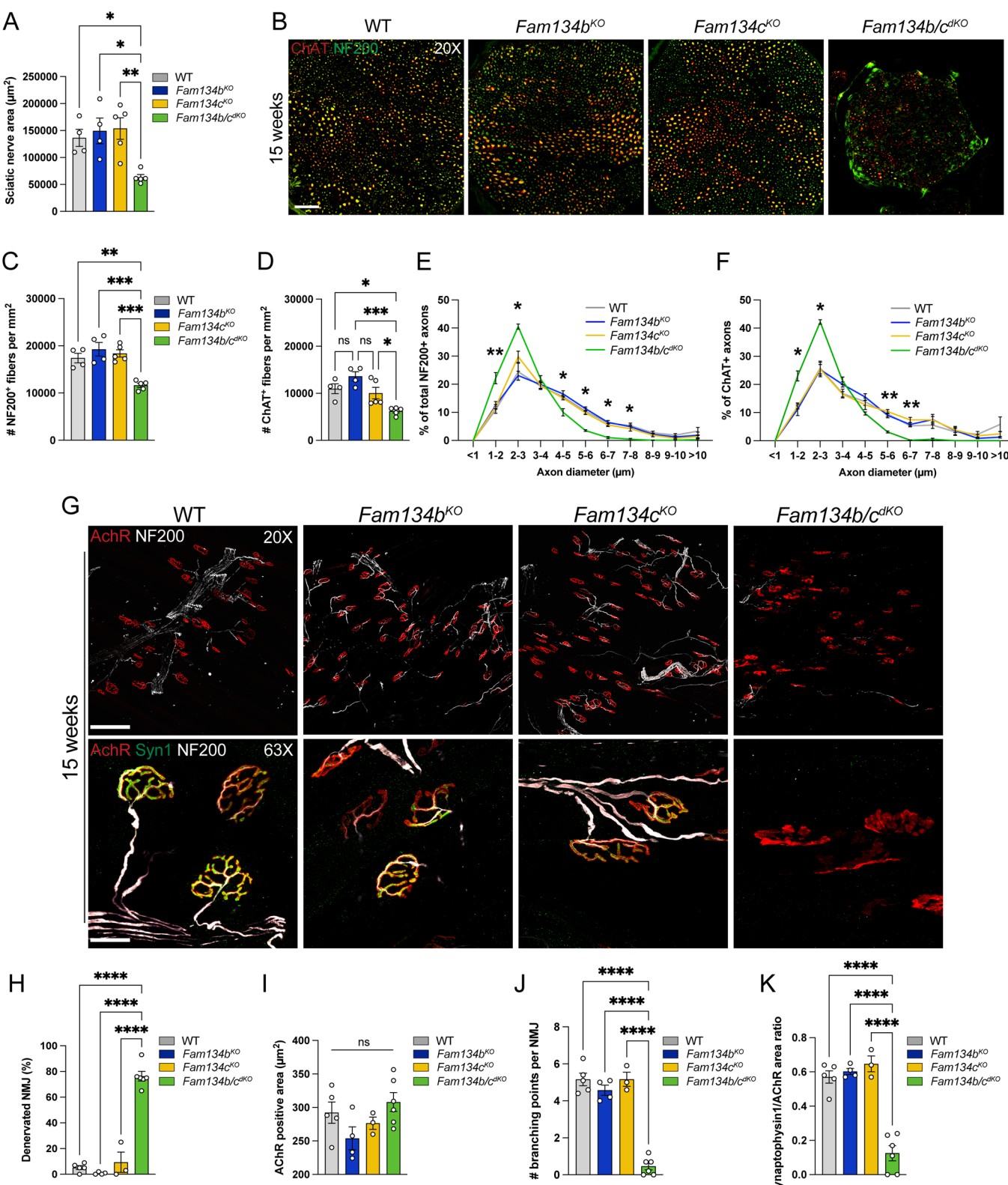

**Figure 3. *Fam134b/c^dKO* leads to axonal neurodegeneration and denervation of neuromuscular junctions.**

(A) Area quantification of tibial sciatic nerve showing a significant decrease in *Fam134b/c^dKO* compared with the other genotypes. (B) Representative immunofluorescence staining of ChAT (red) and NF200 (green) in tibial sciatic nerve sections from WT, *Fam134b^KO*, *Fam134c^KO*, and *Fam134b/c^dKO* mice aged 15 weeks. Scale bar, 50 μm. (C, D) Number of NF200$^+$ (C) or ChAT$^+$ (D) axons per mm$^2$ indicating a decreased density in *Fam134b/c^dKO* compared with the other genotypes. (E, F) Diameter distribution of NF200$^+$ (E) or ChAT$^+$ (F) axons showing accumulation of smaller sized axons in *Fam134b/c^dKO* compared to the other genotypes. (G) Representative immunofluorescence staining of AchR (red), NF200 (gray), and Syn1 (green) in EDL muscle of WT, *Fam134b^KO*, *Fam134c^KO*, and *Fam134b/c^dKO* mice aged 15 weeks. Scale bars, 100 μm and 25 μm, respectively, in the ×20 and ×63 magnification. (H) *Fam134b/c^dKO* muscles show an increased percentage of denervated neuromuscular junctions (NMJs) compared with the other genotypes. (I) AchR mean positive area showed there were no significant changes among the different genotypes. (J) *Fam134b/c^dKO* NMJs show a decreased number of branches compared with the other genotypes. (K) The ratio between Syn1 and AChR positive area is reduced in *Fam134b/c^dKO* NMJs compared with the other genotypes. Data information: $n \geq 4$ animals/group in all experiments. Statistical significance was determined by one-way ANOVA (A, C, D, H–K) or two-way ANOVA (E, F) followed by Tukey's multiple comparisons test. Data represent mean ± SEM. ns $P > 0.05$, *$P < 0.05$, **$P < 0.01$, ***$P < 0.001$, ****$P < 0.0001$. Statistical analysis and exact $P$ values are included in the source data files. Source data are available online for this figure.

The ER is also an important hub for lipid metabolism and, indeed, GO terms related to lipid biosynthetic processes were also found to be altered, including diverse enzymes involved in cholesterol metabolism that were downregulated (such as Fdft1, Dhcr24, Hmgcr, Cyb5r3; Fig. 5D,E).

The lipidome profile in WT, *Fam134b^KO*, *Fam134c^KO*, and *Fam134b/c^dKO* sciatic nerves at 2 weeks showed approximately 400 annotated lipids across all experimental groups. Notably, 67 annotated lipids displayed significant downregulation in *Fam134b/c^dKO* compared with all other groups (Fig. EV5A; Dataset EV2). *Fam134b/c^dKO* lipid alterations encompassed various phospholipid subclasses, including phosphatidylcholine, phosphatidylethanolamine, phosphatidylinositol, and phosphatidylserine (Fig. EV5B), which have pivotal roles not only in the maintenance of membrane integrity but also in cell signaling and cellular metabolic function (Blusztajn et al, 1987).

Overall, our proteomic and lipidomic profiles of *Fam134b/c^dKO* indicate a critical role of Fam134b and Fam134c proteins in maintaining several important functions for axonal health, ranging from cytoskeletal organization, organelle trafficking, synaptic transmission, and lipid metabolism.

The dynamics of ER within axons are crucial for neuronal integrity (Yperman and Kuijpers, 2023). Tubular ER and microtubules regulate neuronal polarity and axonal maintenance (Farias et al, 2019). Our study highlights the significant role of FAM134B and FAM134C in regulating tubular ER homeostasis in mouse axons. Phenotypes observed in *Fam134b/c^dKO* mice, such as reduced body weight, muscle mass, skeletal growth, lifespan, and sensorimotor function, indicate redundant roles of these proteins.

Despite the known effects of HSAN2 in humans, the early-onset and progressive axonal degeneration in the sciatic nerve of *Fam134b/c^dKO* mice affects both sensory and motor fibers. This suggests that FAM134C plays a role in preserving motor functionality in patients with FAM134B mutations. Notably, FAM134C mutations have not been associated yet either to motor or sensory disease susceptibility in humans. However, given the *Fam134b^KO* mouse model shows a late onset of the sensory disease (Khaminets et al, 2015), it is likely that the *Fam134c^KO* mouse model will also develop a phenotype in older mice. The early-onset and rapid progression of the *Fam134b/c^dKO* phenotype suggest that Fam134c may delay or mitigate the severity of HSAN2 axonopathy in mice. This observation opens the possibility of targeting FAM134C activation as a therapeutic strategy for HSAN2.

Downregulation of tubular-ER-shaping proteins Reep1 and Reep2 in *Fam134b/c^dKO* sciatic nerve provides crucial molecular insights into the functions of FAM134B and FAM134C in maintaining tubular ER homeostasis in axons. Loss-of-function mutations of Reep1 and Reep2 have been previously associated with neurodegenerative disorders, such as HSP (Beetz et al, 2013; Esteves et al, 2014). Our data suggest that their altered levels might be implicated in HSAN2 pathogenesis. Interestingly, a different organization of the ER found in *Fam134b/c^dKO* axons from the sciatic nerve was accompanied by altered cytoskeletal components in some axons. As findings reported previously in Atl1$^{KI/KI}$/Reep1$^{-/-}$ and Arl6ip1 KO mice (Foronda et al, 2023; Zhu et al, 2022), we observed that in *Fam134b/c^dKO* mice, instead of running parallel to the longitudinal axonal axis, the ER was organized transversely. These ladder-like ER structures have been previously described in vitro as a feature that becomes less frequent in the axons of mature neurons (Zamponi et al, 2022). Given these observations, it is plausible that *Fam134b/c^dKO* axonal degeneration may be a consequence of an impaired ER reshaping during axonal growth.

The different ER organization in *Fam134b/c^dKO* axons is accompanied by deregulation of various neurofilaments, as well as cytoskeletal components. These findings indicate that both FAM134B and FAM134C are necessary to mature a proper axonal ER shape and, consequently, cytoskeleton integrity and organelle trafficking.

One outstanding question is whether the *Fam134b/c^dKO* phenotype solely results from impaired ER-phagy. Deletion of ER-phagy receptors in iNeurons mimics the ER axonal phenotype seen in iNeurons lacking ATG12 (Hoyer et al, 2024), implying ER-phagy defects as a major cause. Notably, in this study, the individual deletion of an ER-phagy receptor in iNeurons did not induce an axonal phenotype, consistent with functional redundancy among ER-phagy receptors. However, some observations challenge the link to autophagy defects. For instance, reduced Reep proteins in *Fam134b/c^dKO* axons suggest non-autophagy pathways of degeneration. Additionally, disruptions in ER orientation and cytoskeletal organization may not be directly linked to autophagy dysfunction but rather to imbalanced ER-shaping protein proteostasis. Resolving these complexities will require further investigation.

Our in vivo findings shed new light on the critical and redundant roles played by FAM134B and FAM134C in mice, demonstrating their concomitant importance in sustaining the health of axons. Most notably, our data pinpoint the functional relevance of FAM134B and FAM134C to the regulation of axonal tubular but not cortical ER, as evidenced by the fact that the ER of dorsal root ganglion (DRG) sensory neurons and lower motoneurons remains unaffected in *Fam134b/c^dKO* mice, at least up to 4 weeks of age.

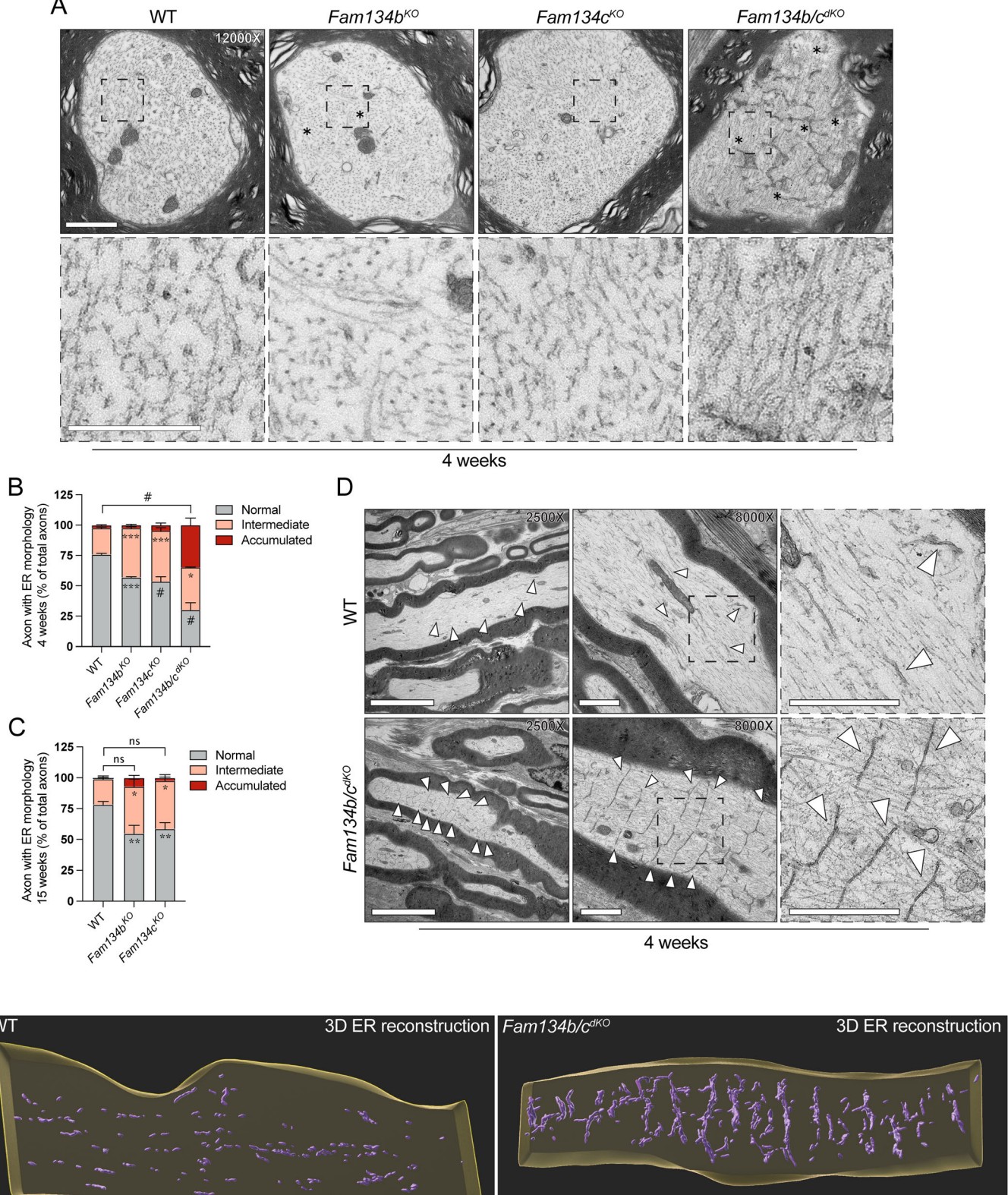

**Figure 4. Early accumulation of ladder-like ER in *Fam134b/c^{dKO}* sciatic nerve axons.**

(A) Representative electron micrographs of ER in tibial nerves axons from WT, *Fam134b^{KO}*, *Fam134c^{KO}*, and *Fam134b/c^{dKO}* mice aged 4 weeks. Alterations of the cytoskeleton are highlighted with asterisks. Scale bar, 1 μm or 500 nm in the insets. (B, C) Analysis of accumulated ER showing a high accumulation in *Fam134b/c^{dKO}* axons at 4 weeks (B), whereas *Fam134b^{KO}* and *Fam134c^{KO}* axons exhibit an intermediate accumulation that is constant at 4 and 15 weeks of age (B, C). *Fam134b/c^{dKO}* axons are not represented in (C) because they were degenerating and thus not analyzed. $n = 3$ animals/group. Statistical significance was determined by two-way ANOVA followed by Tukey's multiple comparisons test. Data represent mean ± SEM. Statistics indicate comparison with WT. ns $P > 0.05$, $*P < 0.05$, $**P < 0.01$, $***P < 0.001$, $^{\#}P < 0.0001$. Statistical analysis and exact $P$ values are included in the source data files. (D) Representative electron micrographs of longitudinal axon tomograms from WT and *Fam134b/c^{dKO}* mice aged 4 weeks. ER membranes are indicated with white arrowheads. Scale bar, 5 μm (×2500 magnifications) and 1 μm (×8000 magnifications and their enlargements). (E) 3D reconstruction of the tomogram in (D) showing ER segmented of WT and *Fam134b/c^{dKO}* mice aged 4 weeks. Scale bar, 0.5 μm. Source data are available online for this figure.

To date, we have no information about the physiological role of FAM134A, and future studies are needed to clarify the relevance of this protein in different tissues. These mouse models represent valuable tools to study the contribution of FAM134 proteins in regulating ER functions in different tissues, uncovering novel physiologically relevant functions of each family member and, more broadly, of ER-phagy.

## Methods

### Animal housing

All mice in this study were housed under pathogen-free conditions at 22 °C and with 12-h dark/12-h light cycles (light cycle from 8:00 a.m. to 8:00 p.m.). Mice were fed with a standard chow diet and were maintained in C57BL6/J strain background. All mice were observed weekly by trained personnel. All studies on mice were conducted in strict accordance with the Institutional Guidelines for animal research and approved by the Italian Ministry of Health (approval no. 154/2022-PR, and n. 68/2023-PR), Department of Public Health, Animal Health, Nutrition, and Food Safety in accordance with the law on animal experimentation (D.Lgs. 26/2014).

### Generation of mouse models and genotyping

Mouse model of *Retreg2* deletion (Fam134a KO) was generated by the Institute of Molecular Genetics of the Czech Academy of Sciences (Prague, Czech Republic). Briefly, a CRISPR strategy was used to obtain exon 2 deletion of Retreg2 in mouse embryonic stem cells before embryo transfer in C57BL/6 WT females. Primers for PCR genotyping encompass the deleted region and produce a 521 bp product for the wild-type allele and a 320 bp product for the mutant allele.

Mouse line with *Retreg1* deletion (Fam134b KO) was generated by Cyagen (California, USA) co-injecting Cas9 mRNA and specific gRNA, generated by in vitro transcription, into fertilized mouse eggs. F0 founder animal sequence analysis showed the deletion of the entire *Retreg1* gene and they normally transmitted the mutation to F1 offspring. The mouse genotyping strategy consisted of a three-primer PCR, with forward and reverse primers encompassing the entire gene sequence and a second reverse primer (defined as R1_WT/HE_Rv in the primer list) targeting the beginning of the deleted region. PCR generated a 720 bp product (for the deleted allele) and a 486 bp product for the wild-type allele.

Mice carrying *Retreg3* deletion (Fam134c KO) were generated using the European Conditional Mouse Mutagenesis/KnockOut Mouse Project (EUCOMM/KOMP)-CSD "Knockout-First" ES cell resource at MRC Harwell Institute (Oxfordshire, UK). The targeting cassette of the transgenic Retreg3 allele contains a promotorless β-galactosidase gene and a neomycin resistance gene flanked by [LoxP-flippase (FLP) recognition target] and LoxP sites flanking exon 2 of the Retreg3 gene. Excision of exon 2 was achieved by crossing the transgenic mice carrying the transgenic cassette with germline Cre-expressing mice. Three primers were used for PCR genotyping; a universal mutant reverse primer (5mut-R1) that sits in the sequence just after the 5' homology arm; a forward primer (5arm-WTF) targeting the 5' homology arm and designed to give the mutant-specific band of 119 bp that will only be present if the cassette is present; a WT reverse primer (Crit-WTR) designed to the critical region that gives a product of 269 bp. If the mutant cassette is present, the product between the forward primer and the Crit-WTR reverse primer is too large to be amplified by PCR under standard conditions.

*Fam134a/b^{dKO}*, *Fam134a/c^{dKO}*, and *Fam134b/c^{dKO}* mouse models were generated by crossing the respective single KO mouse models described above.

All the primers described in the PCR genotyping strategy are listed in Table 2.

### Behavioral procedures

#### Hindlimb clasping test

Mice were suspended by their tail and the extent of hindlimb clasping was observed for 30 sec. If both hindlimbs were splayed outward away from the abdomen with splayed toes, a score of "0" was given. If one hindlimb was retracted or both hindlimbs were partially retracted toward the abdomen without touching it and the toes were splayed, a score of "1" was assigned. If both hindlimbs were partially retracted toward the abdomen and were touching the abdomen without touching each other, a score of "2" was given. If both hindlimbs were fully clasped and touching the abdomen, a score of "3" was assigned (Guyenet et al, 2010).

#### Open field test

All animals were left free to explore a Plexiglas arena (35 × 47 × 60 cm) for 20 min. The distance traveled (m) and maximum speed (m/s) were automatically scored using a video camera (PANASONIC WV-BP330) connected to a video-tracking system (ANY-MAZE, Stoelting, USA).

#### The hanging wire and hanging steel test

Neuromuscular strength was assessed by recording the latency (s) to fall down from a wire turned upside-down in the Hanging wire

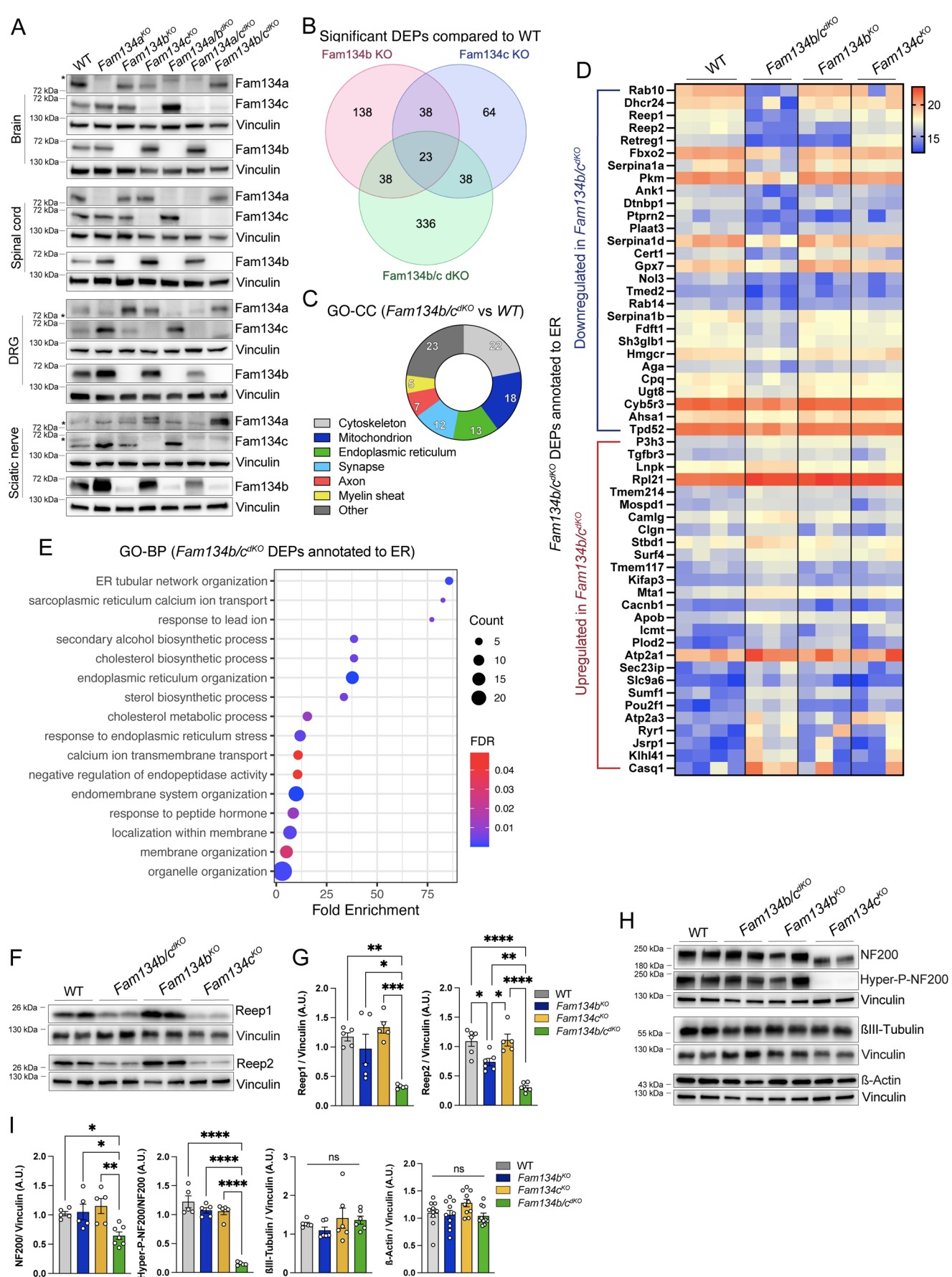

**Figure 5. Proteins related to the organization of tubular ER and cytoskeleton are reduced in *Fam134b/c^dKO^* sciatic nerves.**

(A) Western blot analysis showing protein expression of Fam134 proteins in brain, lumbar spinal cord, lumbar DRG, and sciatic nerve of 2-week-old mice with the indicated genotypes. Asterisks indicate a specific band. (B) Venn diagram showing significant differentially expressed proteins (DEPs) in sciatic nerves of *Fam134b^KO^*, *Fam134c^KO^*, and *Fam134b/c^dKO^* compared with WT. (C) Cellular components gene ontology enrichment in *Fam134b/c^dKO^* DEPs are represented as a percentage of total *Fam134b/c^dKO^* DEPs. (D) Heatmap based on significant *Fam134b/c^dKO^* DEPs annotated to ER and comparing WT, *Fam134b^KO^*, and *Fam134c^KO^*, and *Fam134b/c^dKO^* protein intensity values. The protein abundance scale is depicted on the top right. (E) Most relevant terms from Gene Ontology Biological Process (GO-BP) of *Fam134b/c^dKO^* DEPs annotated to ER are represented by dot plots indicating FDR, DEP count, and fold enrichment. $n \geq 3$ animals/group. FDR < 0.05. (F) Western blot analysis showing Reep1 and Reep2 protein levels in the sciatic nerve of 2-week-old mice with the indicated genotypes. (G) Quantification of Reep1 and Reep2 protein levels normalized to vinculin. $n \geq 5$ animals/group. (H) Western blot analysis showing total and hyperphospho- neurofilament heavy chain (NF200), beta-III-Tubulin, and beta-Actin protein levels in the sciatic nerve of 2-week-old mice with the indicated genotypes. (I) Protein quantification relative to (H) normalized to vinculin, or to total NF200 in the case of Hyperphospho-NF200. $n \geq 5$ animals/group. Data information: Statistical significance was determined by one-way ANOVA (G, I) followed by Tukey's multiple comparisons test. Data represent mean ± SEM. ns $P > 0.05$, *$P < 0.05$, **$P < 0.01$, ***$P < 0.001$, ****$P < 0.0001$. Statistical analysis and exact $P$ values are included in the source data files. Source data are available online for this figure.

**Table 2. Used primers for genotyping.**

| Primer name | Sequence (5'→3') |
| --- | --- |
| 5arm-WTF | GGCATTAATATACACAATAGCACAA |
| Crit-WTR | GGCACGTGGATTTCTGAGTT |
| 5mut-R1 | GAACTTCGGAATAGGAACTTCG |
| R1_Fw | AAGCGACTTTCACCAGCGTGTG |
| R1_Rv | GACGATAACACAATACAGAAACATCGG |
| R1_WT/HE_Rv | GAGGGTGCGGTCAAGTTCGTG |
| R2_Fw | CACAGCGACTGCTGGTTTGG |
| R2_Rv | GGATGCCTTAGGAGATAAATGG |

test and from an elevated horizontal support in the hanging steel test. The cut-off time was 120 s.

### Rotarod motor test

Motor capabilities were evaluated on a Rotarod machine, under an accelerating protocol. Animals underwent a four-trial test under an accelerating protocol going from 4 rpm to 40 rpm in 5 min for 2 days, with an intertrial rest of 20 min. The latency is presented as the average of the total trials.

### Hot plate test

The hot plate test was performed on a warm plate at 52 °C. The response time to observed behavioral parameters, such as paw licking and stamping, was timed for three trials, separated by a 30-minute break. The cut-off time was 30 s.

## Extracellular single-unit recording in vivo

For in vivo single-unit electrophysiological recordings, the mice were first anesthetized with sodium pentobarbital (50 mg/kg i.p.) and then underwent maintenance anesthesia with propofol (5–10 mg/kg/h, i.v.) after incannulating the jugular vein and then inserting a catheter necessary for the slow infusion of the anesthetic. Once the depth of anesthesia had been verified, after making an incision of the skin and dorsal muscles in order to expose the vertebral segments between L4-L6, a laminectomy was then performed without damaging the spinal cord. The animals were then attached to a stereotaxic apparatus (David Kopf Instruments, Tujunga, CA, USA) and supported by clamps attached to the vertebral processes on either side of the exposure site. Body temperature was maintained around 37 °C by the use of a

temperature-controlled heated pad. After the spinal application of mineral oil, a tungsten recording electrode (3–5 MΩ; FHC Frederick Haer & Co., Bowdoin, ME, USA) was slowly lowered into the dorsal horn of the spinal cord to record the activity of specific nociceptive neurons in the dorsal horn. In order to isolate a neuron, the electrode was moved slowly along the segments of interest while the hind paw (receptive field) was mechanically stimulated for 2–3 s through a von Frey filament with a bending force of 97.8 mN (noxious stimulation) for 2 s until it deformed slightly. Only neurons responding to mechanical stimulation of the paw were included in the study. Once the neuron was identified, stimulation was repeated every 300 s. The recorded signals were then amplified and displayed on a digital memory oscilloscope to ensure that the unit under study was unambiguously discriminated throughout the experiment. The signals were also sent to a window discriminator and then processed by a CED 1401 interface (Cambridge Electronic Design Ltd., Milton, UK) connected to a Pentium III PC. Spike2 software (CED, version 4) was used to create histograms of the peristimulus frequency online and to store and analyze digital recordings of the activity of individual units offline. The configuration, shape, and height of the recorded action potentials were monitored and recorded continuously using a window discriminator and Spike2 software for online and offline analysis. This study included only those neurons whose spike configuration remained constant and could be clearly discriminated from the activity in the background during the experiment, indicating that the activity of only one neuron and the same neuron was measured. Neuronal activity was expressed as spikes/s (Hz). At the end of the experiment, each animal was sacrificed with a lethal dose of urethane.

## Western blotting

All tissues were lysed using RIPA buffer containing protease and phosphatase inhibitor cocktails. The brain was homogenized using a Tissue Lyser II (QIAGEN) and left on a rotating wheel overnight at 4 °C. Lumbar dorsal root ganglia, lumbar spinal cord, and sciatic nerve were homogenized by using a motorized pellet pestle (Sigma). Then, samples were sonicated using Bioruptor PLUS (Biosense; 20 s ON, 10 s OFF, 10 cycles, at 4 °C), and pellets were discarded by spinning down for 15 min at maximum speed. BCA assay was performed on supernatants to determine protein concentration.

All antibodies used are listed in Table 3.

**Table 3. Detailed information for antibodies and reagents.**

| Reagent/resource | Source | Catalog number | Dilution | Technique |
|---|---|---|---|---|
| **Antibodies** | | | | |
| ChAT (goat) | EMD Millipore | AB144P | 1:100 | IF |
| NF200 (rabbit) | SySy | 171102 | 1:200/ 1:1000 | IF/WB |
| Synaptophysin 1 (chicken) | SySy | 101006 | 1:100 | IF |
| Fam134a (rabbit) | Sigma | HPA011844 | 1:1000 | WB |
| Fam134b (rabbit) | Sigma | HPA012077 | 1:1000 | WB |
| Fam134c (rabbit) | Sigma | HPA016492 | 1:1000 | WB |
| Vinculin (mouse) | Sigma | V9264-100UL | 1:5000 | WB |
| Reep1 | Proteintech | 17988-1-AP | 1:1000 | WB |
| Reep2 | Proteintech | 15684-1-AP | 1:1000 | WB |
| SMI31 | Biolegend | 801601 | 1:1000 | WB |
| Beta-III-tubulin | Thermo Scientific | MA1-118 | 1:2000 | WB |
| Beta-actin | Novus Biologicals | NB600-501 | 1:2000 | WB |
| Calnexin | Enzo Life Sciences | ADI-SPA-860-D | 1:1000 | WB |
| Atl3 | Proteintech | 16921-1-AP | 1:1000 | WB |
| Ccpg1 | Proteintech | 13861-1-AP | 1:1000 | WB |
| Tex264 | Novus Biologicals | NBP1-89866 | 1:1000 | WB |
| Rtn3 | Proteintech | 12055-2-AP | 1:1000 | WB |
| Sqstm1/p62 | Abnova | H00008878-M01 | 1:1000 | WB |
| LC3b | Novus Biologicals | NB100-2220 | 1:1000 | WB |
| NeuN (mouse) | EMB Millipore | MAB377 | 1:500 | IF |
| CGRP (rabbit) | Sigma | C8198-25UL | 1:100 | IF |
| NF200 (mouse) | Sigma | N0142-100UL | 1:200 | IF |
| GFAP | Sigma | G3893 | 1:200 | IF |
| Iba1 | Novus Biologicals | NB100-1028 | 1:200 | IF |
| MAP2 | SySy | 188 004 | 1:200 | IF |
| Donkey α-Goat 568 | Thermo Scientific | A11057 | 1:500 | IF |
| Donkey α-rabbit 488 | Thermo Scientific | A21206 | 1:500 | IF |
| Goat α-rabbit 568 | Thermo Scientific | A11011 | 1:500 | IF |
| Goat α-mouse 647 | Thermo Scientific | A21235 | 1:500 | IF |
| Goat α-chicken 488 | Thermo Scientific | A21467 | 1:500 | IF |
| Goat α-mouse HRP | Proteintech | SA00001-1 | 1:2000 | WB |
| Goat α-rabbit HRP | Vector Lab | PI-1000-1 | 1:2000 | WB |

**Table 3.** (continued)

| Reagent/resource | Source | Catalog number | Dilution | Technique |
|---|---|---|---|---|
| **Reagents** | | | | |
| Bungarotoxin 568 | Biotium | 00006-100ug | 1:1000 | IF |
| Isolectin B4 (biotin-conjugated) | Sigma | L2140 | 1:100 | IF |
| Streptavidin 488 | Thermo Scientific | S11223 | 1:1000 | IF |

## Immunofluorescence assays

For immunofluorescence assays, anesthetized mice (ketamine and medetomidine) were perfused transcardially with 0.9% saline solution followed by 4% paraformaldehyde (PFA) solution. Brain, sciatic nerve, lumbar dorsal root ganglia (DRG), and EDL muscle were incubated overnight in 4% PFA at 4 °C. Immunostaining of brain, sciatic nerve, and DRG was performed on 10-μm-thick OCT-embedded sections. Sections were incubated in 10% (w/v) goat serum, and 0.3% Triton X-100 in PBS for 1 h. Primary antibodies were incubated overnight at 4 °C. Sections were washed three times with 0.1% BSA in PBS and then incubated for 1 h with secondary antibodies Alexa Fluor–conjugated and 4′,6-diamidino-2-phenylindole (DAPI). Then, sections were mounted in the Mowiol mounting medium. For EDL immunostaining, the muscle was washed in PBS and cut longitudinally into 6/8 pieces. Tissue was blocked in 3% BSA, 5% goat serum, and 0.5% Triton X-100 in a 2-mL tube on a rotating wheel for 2 h and incubated with primary antibodies overnight on the rotating wheel at 4 °C. After PBS washing tissue was incubated with secondary antibodies Alexa Fluor–conjugated for 2 h on the rotating wheel at room temperature. After PBS washing, tissue was mounted on a slide using Mowiol medium and gently pressed with the coverslip. Fluorescence images were acquired with an LSM 880 confocal microscope (Carl Zeiss) or an Eclipse Ti2-E (Nikon) equipped with a 10×, 20×, and 63× objective and using a slice thickness of 0.5 μm for the z-stack. All the quantifications were performed using ImageJ (National Institutes of Health, Bethesda). For axon and NMJ quantification, images were processed through the threshold tool of the ImageJ software to segment out and measure the staining areas. Axon number and size were calculated by using the Analyze Particles tool of the ImageJ software, excluding signals smaller than $1\ \mu m^2$.

Automated immunofluorescence standardized assays were performed in 20-μm cryosections with VENTANA BenchMark Ultra automated staining instrument (Ventana Medical Systems, Roche), using VENTANA reagents according to the manufacturer's instructions. Epitope retrieval was accomplished with CC1 solution (cat # 950–224) at a high temperature (95 °C). Slides were developed using DISCOVERY UltraMap anti-Gt HRP (RUO; Roche 760-4648) and DISCOVERY UltraMap anti-Ms HRP (RUO; Roche 760-4313) according to the manufacturer's instructions. Nuclei were stained with DAPI (Thermo Fisher Scientific, D1306).

All antibodies used are listed in Table 3.

## Semithin sections and electron microscopy

Sciatic nerves and spinal cords were postfixed in 2% glutaraldehyde (Sigma) in 0.1 M cacodylate buffer or 2% glutaraldehyde (Sigma) and 2% PFA (Life Tech) in 0.12 M PB buffer for 48 h. Then tissues were treated with 1% osmium tetroxide (Sigma), dehydrated using ethanol and propylene oxide, and embedded in Epon resin (Sigma). Semithin cross-sections of 900 nm were cut using an ultramicrotome (EM UC7, Leica) and stained with 1% toluidine blue. Images of sections were scanned using an automated slidescanner (Hamamatsu S360). Degenerating axons were identified as dark spots, manually counted, and normalized by the area of the section. Myelinated axons were automatically segmented and quantified using the Trainable Weeka Segmentation plugin of ImageJ (National Institutes of Health, Bethesda); axons smaller than $5 \mu m^2$ were not considered. For electron microscopy, ultrathin sections of 70 nm were cut from Epon blocks and stained with uranyl acetate (Plano GMBH) and lead citrate (Sigma). Then, images were acquired using a transmission electron microscope (TEM, JEM-2100 Plus, Jeol) equipped with a Gatan ONE View camera. The morphology of the ER was manually evaluated by an experimenter blind to the experiment and classified as normal, intermediate, and accumulated as exemplified in Appendix Fig. S7A. The g-ratio of axons was obtained by dividing the inner axon diameter by the outer myelin diameter. For electron tomography, 300 nm longitudinal sections were cut using an ultramicrotome (Leica Microsystems, UC6) and a diamond knife (Science Services # DU3530) and mounted onto 200 mesh copper grids without film. Grids were stained from both sides with 1.5% uranyl acetate (Agar Scientific, # R1260A) for 20 min and 3% Reynolds lead citrate solution made from Lead (II) nitrate (Roth, # HN32.1) and tri-Sodium citrate dehydrate (Roth #4088.3) for 3 min. Images were acquired using a JEM-2100 Plus Transmission Electron Microscope (JEOL) operating at 200 kV equipped with a OneView 4 K camera (Gatan). Tilt series from the same ROI in two consecutive 300 nm sections were acquired at 5000X magnification (pixel size: 2.193 nm) using SerialEM (Mastronarde, 2005) and reconstructed and registered on top of each other using IMOD (Kremer, 1996). ER was identified by distinguishable and enclosed membranes in at least one image of the stack and segmented using Microscopy imaging Browser (Belevich et al, 2016) and visualized using IMARIS (Oxford Instruments).

## Proteomic analysis

Both sciatic nerves from each mouse were minced in small pieces and digested in 150 μL of RIPA buffer supplemented with protease and phosphatase cocktail inhibitors. Samples were mechanically disrupted using a motorized pestle mixer and then sonicated for 15 cycles (30 s ON, 30 s OFF at 4 °C and low intensity) using the Bioruptor Plus® device (Diagenode). Then, pellets were discarded by spinning down for 30 min at maximum speed, and a BCA assay was performed on supernatants to determine protein concentration.

For proteomic analysis, the equivalent of 30 μg of sciatic nerve lysate was precipitated by methanol/chloroform using four volumes of ice-cold methanol, one volume of chloroform, and three volumes of water. The mixture was centrifuged at $20,000 \times g$ for 30 min, the upper aqueous phase was removed, and three volumes of ice-cold methanol were added. Proteins were pelleted by centrifugation and washed twice with one volume of ice-cold methanol and air-dried. The resulting protein pellet was resuspended in 20 μL of GnHCl buffer (6 M Guanidine hydrochloride, 50 nM Tris pH 8.5, 5 mM TCEP, 20 mM CAA) and boiled at 95 °C for 10 min. For digestion, proteins were diluted 1:2 and incubated 1:100 (w/w) ratio with a mixture of LysC and sequencing-grade trypsin (Thermo) overnight agitating at 37 °C. The reaction was acidified using trifluoroacetic acid (TFA) (0.5%) and purified using Sep-Pak tC18 (Waters, 50 mg) according to the manufacturer's protocol.

Instruments for LC-MS/MS analysis consisted of NanoLC 1200 coupled via a nanoelectrospray ionization source to the quadrupole-based Q Exactive HF benchtop mass spectrometer. Peptide separation was carried out according to their hydrophobicity on a homemade chromatographic column, 75-μm inside diameter, 8-μm tip, bed-packed with Reprosil-PUR (C18-AQ), 1.9-μm particle size, 120-Å pore size, using a binary buffer system consisting of solution A (0.1% formic acid) and solution B (80% acetonitrile and 0.1% formic acid). Runs of 120 min after loading were used for proteome samples, with a constant flow rate of 300 nl/min. After sample loading, the run started at 5% buffer B for 5 min, followed by a series of linear gradients, from 5 to 30% B in 90 min, then a 10-min step to reach 50% and a 5-min step to reach 95%. This last step was maintained for 10 min. Q Exactive HF settings are as follows: MS spectra were acquired using 3E6 as an AGC target, a maximal injection time of 20 ms, and a 120,000 resolution at 200 mass/charge ratio (m/z). The mass spectrometer was operated in a data-dependent Top20 mode with subsequent acquisition of higher-energy collisional dissociation fragmentation MS/MS spectra of the top 20 most intense peaks. Resolution, for MS/MS spectra, was set to 15,000 at 200 m/z, AGC target to 1E5, maximum injection time to 20 ms, and the isolation window to 1.6 Th. The intensity threshold was set at 2.0 E4 and dynamic exclusion at 30 s.

For statistical analysis, the Perseus software (1.6.2.3) was used to logarithmize, group and filter [after filter] the protein abundance. ANOVA and *t*-test analysis were performed setting FDR = 0.05. Proteins with ANOVA *q* value or Log2 Difference $\geq \pm 1$ and *q* value < 0.05 were considered significantly enriched.

## Lipidomic analysis

Lipids were extracted from tissue homogenates as follows: 100 μL of ice-cold MeOH/H$_2$O (1:1 v/v %) was added to the samples that were shacked, put in an ultrasound bath and subsequently centrifuged for 20 min at 4 °C and 20,238 rcf. Supernatants were discarded and pellets were extracted with 100 μL of ice-cold MTBE containing a mix of deuterated standard and were re-incubated in a Thermomixer (Eppendorf) for 1 h × 4 °C × 550 rpm. Finally, 188 μL of H$_2$O were added to samples and after centrifugation, the upper phases were separated and dried. Supernatants were dried using a SpeedVac (Savant, Thermo Scientific, Milan, Italy). Untargeted lipidomics analyses were performed on an Ultimate RS3000 UHPLC (Thermo Fisher Scientific, Milan, Italy). The LC system was coupled online to a TimsTOF Pro Quadrupole Time of Flight (Q-TOF) (Bruker Daltonics, Bremen, Germany) equipped with an Apollo II electrospray ionization (ESI) probe. Lipid separation was performed with an Acquity UPLC CSHTM C18 column (50 × 2.1 mm; 1.7 μm, 130 Å) protected with a VanGuard CSHTM precolumn (5.0 × 2.1 mm; 1.7 μm, 130 Å) (Waters, Milford, MA, USA). The TIMS-MS analyses

were performed in data-dependent parallel accumulation serial fragmentation (DDA-PASEF) in both positive and negative ionization, in separate runs, each sample was injected in triplicate. Data analysis, including 4D data alignment, filtering, and annotation was performed with MetaboScape 2023b (Bruker) employing a feature-finding algorithm (T-Rex 4D) that automatically extracts buckets from raw files. Lipid annotation was performed first with a rule-based annotation, based on diagnostic-class-specific fragments and their intensity in acquired MS/MS spectra, and, subsequently, using the LipidBlast spectral library of MS DIAL (http://prime.psc.riken.jp/compms/msdial/main.html). CCS values were compared with those predicted by the CCSbase platform (https://ccsbase.net/) and CCS-Predict tool by MetaboScape. The assignment of the molecular formula was performed for the detected features using Smart Formula™ (SF). Each lipid feature was manually curated following Lipidomics Standard Initiative (LSI) guidelines (https://lipidomics-standards-initiative.org/guidelines/lipid-species-identification/general-rules). Detailed RP-UHPLC-TIMS method as well as processing parameters and annotation criteria are reported elsewhere (Merciai et al, 2022). Before statistical analysis data were normalized against class-specific internal standard, log-transformed and autoscaled.

### Statistical analysis

The results were expressed as mean ± SEM. Statistical analyses were performed using Statview 5.0 (SAS Institute Inc., North Carolina, USA) and GraphPad Prism 8 software. The statistical significance was assessed using one-way ANOVA, two-way ANOVA, or repeated measures ANOVA, followed by Tukey's post hoc test if appropriate. Before applying the ANOVA tests, data distribution normality was tested with the Kolmogorov–Smirnov test. Outliers were evaluated with the z-score function.

### Graphics

Synopsis image and images of behavioral tests shown in Figs. 1 and EV1 were generated using BioRender.com.

## Data availability

The mass spectrometry proteomic data have been deposited to the ProteomeXchange Consortium via the PRIDE (Perez-Riverol et al, 2022) partner repository at the following link http://www.ebi.ac.uk/pride/archive/projects/PXD050983. The lipidomic data have been deposited to the Zenodo repository at the following link https://doi.org/10.5281/zenodo.10879359.

The source data of this paper are collected in the following database record: biostudies:S-SCDT-10_1038-S44319-024-00213-7.

## Peer review information

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

## Acknowledgements

The authors thank Cathal Wilson and Graciana Diez Roux (TIGEM) for their suggestions and critical reading of the manuscript. The authors thank advanced microscopy, Mass spectrometry, and advanced histopathology facilities at TIGEM Institute. The authors thank E. Polishchuk for helping us with electron microscopy experiments. The authors thank the CECAD imaging facility for technical assistance. CS acknowledges the Italian Telethon funding agency (GSA21F005), the European Research Council (ERC-101045285-AUTOSELECT), and Italian Ministry of Health (PNRR-MAD-2022-12376672).

## Author contributions

**Francescopaolo Iavarone**: Conceptualization; Data curation; Formal analysis; Validation; Investigation; Visualization; Methodology; Writing—original draft.

**Marta Zaninello**: Data curation; Formal analysis; Investigation; Visualization; Methodology; Writing—review and editing. **Michela Perrone**: Data curation; Formal analysis. **Mariagrazia Monaco**: Data curation; Formal analysis. **Esther Barth**: Formal analysis; Methodology. **Felix Gaedke**: Formal analysis; Methodology. **Maria Teresa Pizzo**: Formal analysis. **Giorgia Di Lorenzo**: Data curation; Formal analysis. **Vincenzo Desiderio**: Resources. **Eduardo Sommella**: Data curation; Formal analysis. **Fabrizio Merciai**: Data curation; Formal analysis. **Emanuela Salviati**: Formal analysis. **Pietro Campiglia**: Formal analysis. **Livio Luongo**: Conceptualization; Writing—review and editing. **Elvira De Leonibus**: Conceptualization; Supervision; Writing—review and editing. **Elena Rugarli**: Conceptualization; Supervision; Writing—review and editing. **Carmine Settembre**: Conceptualization; Resources; Supervision; Funding acquisition; Project administration; Writing—review and editing.

Source data underlying figure panels in this paper may have individual authorship assigned. Where available, figure panel/source data authorship is listed in the following database record: biostudies:S-SCDT-10_1038-S44319-024-00213-7.

## Disclosure and competing interests statement

The authors declare no competing interests.

# Expanded View Figures

**Figure EV1.** *Fam134b/c* **combined deletion determines a progressive neurological impairment.**

(A) *Fam134b/c^dKO* mice show reduced body weight at 15 weeks of age, as compared with the other Fam134 mutant mice. $n \geq 7$ animals/group. (B) Representative images of tricep and gastrocnemius muscles from WT, *Fam134b^KO*, *Fam134c^KO*, and *Fam134b/c^dKO* mice aged 15 weeks. Scale bar, 5 mm. (C) The weight of tricep and gastrocnemius muscles are massively reduced in *Fam134b/c^dKO* mice. $n \geq 4$ animals/group. (D) Representative images of 4-week-old mice with the indicated genotypes showing hindlimb clasping of *Fam134b/c^dKO* mice while they are suspended by the tail. The images relative to WT and *Fam134b/c^dKO* mice are also shown in Fig. 1G. (E, F) Open field test scores relative to total distance (E) and maximum speed (F) performed by WT, *Fam134b^KO*, and *Fam134c^KO*, and *Fam134b/c^dKO* mice aged 4 or 15 weeks, indicating a worsening of performance in 15-week-old *Fam134b/c^dKO* mice compared with the other genotypes. $n \geq 12$ animals/group. (G) Track plot relative to the exploration of WT, *Fam134b^KO*, *Fam134c^KO*, and *Fam134b/c^dKO* mice during the open field test at the indicated age, showing a dramatic decrease of *Fam134b/c^dKO* mice movement at 15 weeks of age. (H) Hot plate test indicating increased latency (s) of first hind paw pain response in *Fam134b/c^dKO* mice aged 4 and 15 weeks respect with the indicated genotypes. $n \geq 12$ animals/group. (I–K) Extracellular single-unit in vivo recordings of spinal nociceptive neurons in 4-week-old mice, measuring the spontaneous activity or firing rate (I), the activity evoked by mechanical stimulation (J), as well as the duration of evoked activity (K). *Fam134b/c^dKO* and, at a lower extent, *Fam134c^KO* mice exhibit an increase in all the parameters analyzed compared with the other genotypes. $n \geq 3$ animals/group. Data information: Statistical significance was determined by one-way ANOVA (A, C, E, F, H–K) followed by Tukey's multiple comparisons test. Data represent mean ± SEM. ns $P > 0.05$, *$P < 0.05$, **$P < 0.01$, ***$P < 0.001$, ****$P < 0.0001$. Statistical analysis and exact $P$ values are included in the source data files. Images of behavioral tests were generated using Biorender. Source data are available online for this figure.

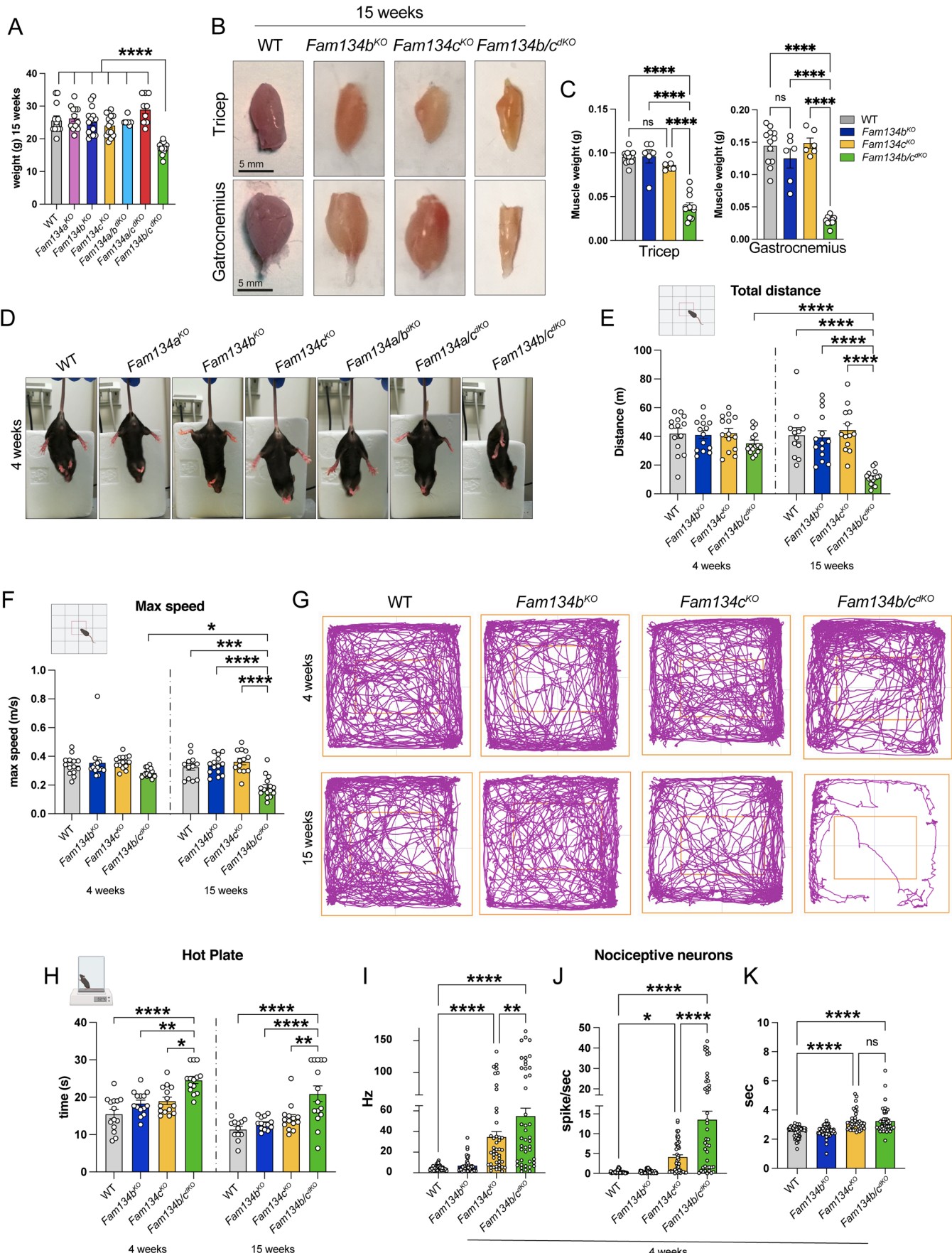

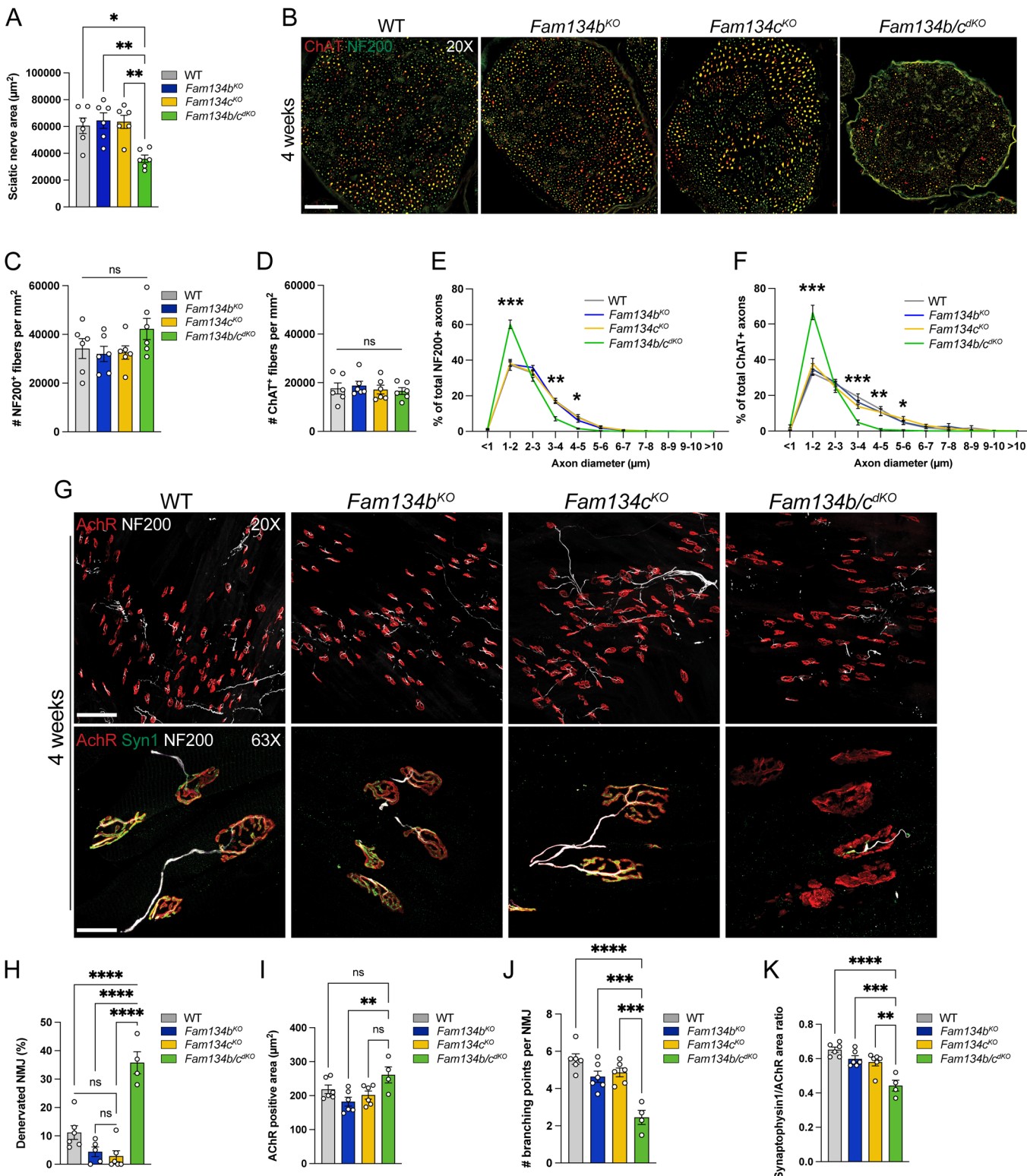

◄ **Figure EV2. Early onset of axonal neurodegeneration and denervation of neuromuscular junctions in *Fam134b/c^dKO*.**

(A) Quantification of the tibial sciatic nerve area shows a significant decrease in *Fam134b/c^dKO* compared with the other genotypes. (B) Representative immunofluorescence staining of ChAT (red) and NF200 (green) in tibial sciatic nerve sections from 4-weeks-old WT, *Fam134b^KO*, *Fam134c^KO* and *Fam134b/c^dKO* mice. Scale bar, 50 µm. (C, D) Number of NF200$^+$ (C) or ChAT$^+$ (D) axons per mm$^2$ showing no difference among genotypes. (E, F) Diameter distribution of NF200$^+$ (E) or ChAT$^+$ (F) axons showing accumulation of smaller sized axons in *Fam134b/c^dKO* compared with the other genotypes. (G) Representative immunofluorescence staining of AchR (red), NF200 (gray), and Syn1 (green) in EDL muscle from 4-weeks-old WT, *Fam134b^KO*, *Fam134c^KO*, and *Fam134b/c^dKO* mice. Scale bars, 100 µm and 25 µm, respectively, in the ×20 and ×63 magnification. (H) *Fam134b/c^dKO* muscles show an increased percentage of denervated neuromuscular junctions (NMJs) compared with the other genotypes. (I) AchR mean positive area indicates no main changes among the different genotypes. (J) *Fam134b/c^dKO* NMJs show a decreased number of branches compared with the other genotypes. (K) The ratio between the Syn1 and AChR positive area is reduced in *Fam134b/c^dKO* NMJs compared to the other genotypes. Data information: $n \geq 4$ animals/group in all experiments. Statistical significance was determined by one-way ANOVA (A, C, D, H–K) or two-way ANOVA (E, F), followed by Tukey's multiple comparisons test. Data represent mean ± SEM. ns $P > 0.05$, *$P < 0.05$, **$P < 0.01$, ***$P < 0.001$, ****$P < 0.0001$. Statistical analysis and exact $P$ values are included in the source data files. Source data are available online for this figure.

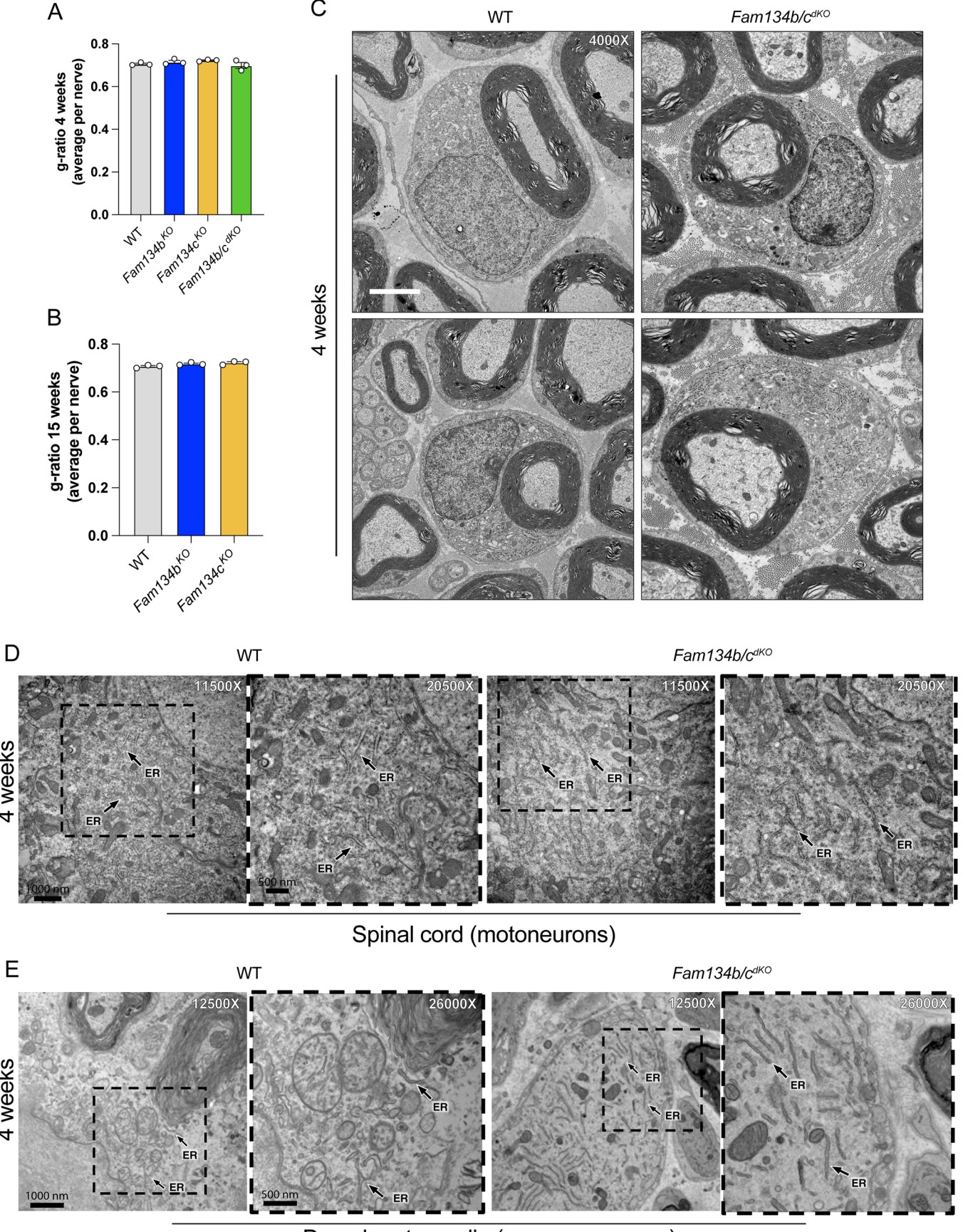

◀

**Figure EV3. Schwann cell, motoneuron, and sensory neuron soma show no ultrastructural alteration in *Fam134b/c^dKO* mice.**

(A, B) Quantification of the g-ratio of WT, *Fam134b^KO*, *Fam134c^KO*, and *Fam134b/c^dKO* myelinated axons from tibial nerves aged 4 (A) and 15 (B) weeks. $n = 3$ animals/group. Data represent mean ± SEM. (C) Representative electron micrographs showing Schwann cells in WT and *Fam134b/c^dKO* sciatic nerve aged 4 weeks. Scale bar, 2 μm. (D) Representative electron micrographs of motoneurons in the ventral horn of the lumbar spinal cord from WT and *Fam134b/c^dKO* mice aged 4 weeks. (E) Representative electron micrographs of neurons in lumbar DRG from WT and *Fam134b/c^dKO* mice aged 4 weeks. Scale bars, 1000 nm or 500 nm in the insets. Source data are available online for this figure.

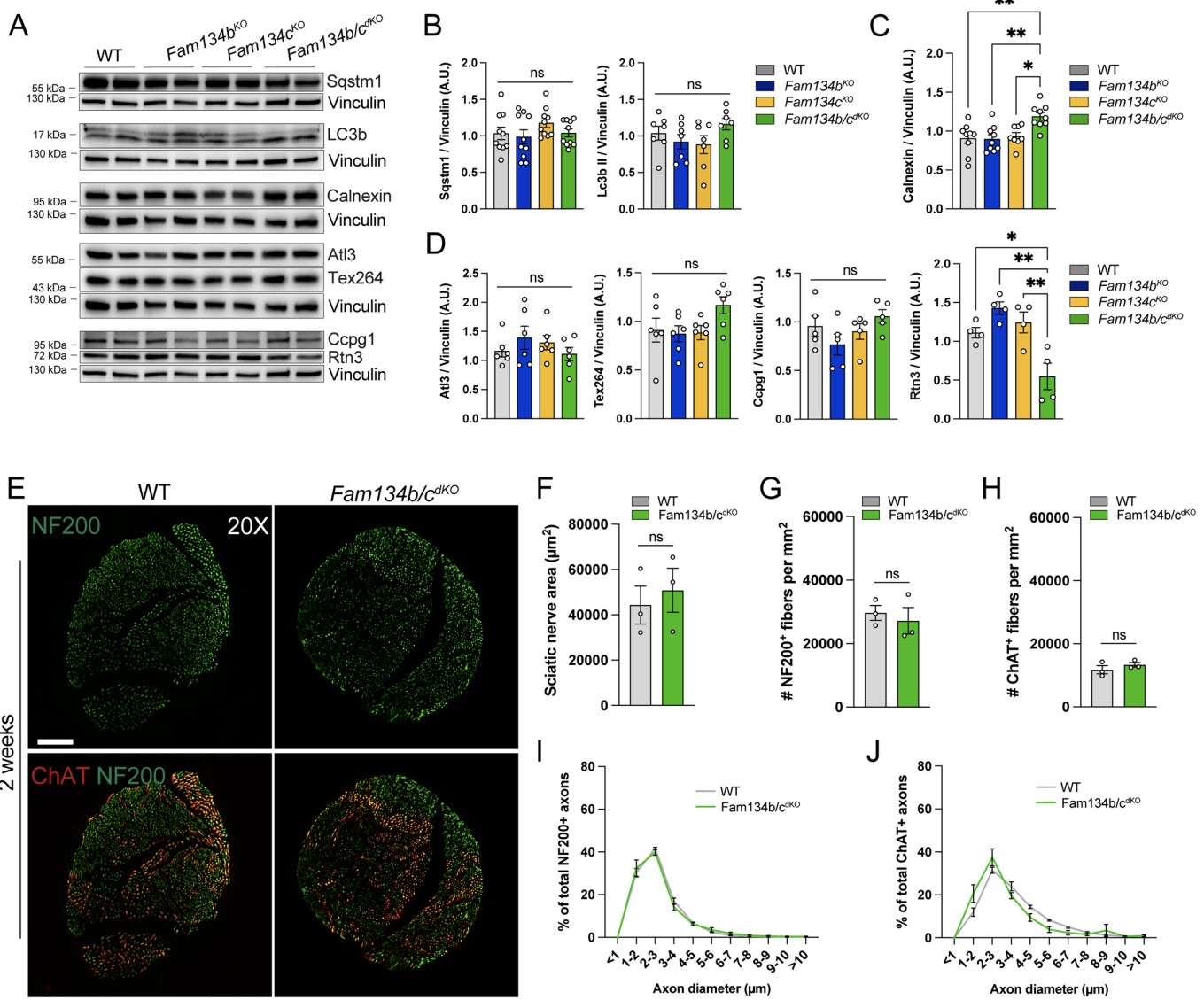

**Figure EV4. Autophagy and histology of sciatic nerve show no major alterations in 2-week-old *Fam134b/c^dKO* mice.**

(A) Western blot analysis showing protein expression of autophagy markers (Sqstm1/p62 and Lc3b), the ER marker Calnexin, and ER-phagy receptors (Atl3, Tex264, Ccpg1, and Rtn3) in sciatic nerve of 2-week-old mice with the indicated genotypes. (B–D) Protein level quantification of autophagy markers (B), Calnexin (C), and ER-phagy receptors (D). Normalized to vinculin. $n \geq 4$ animals/group. (E) Representative immunofluorescence staining of ChAT (red) and NF200 (green) in tibial sciatic nerve sections of WT and *Fam134b/c^dKO* mice aged 2 weeks. Scale bar, 50 µm. (F) Quantification of the tibial sciatic nerve area showing no difference between WT and *Fam134b/c^dKO* mice. (G–J) Morphometric analysis of axons shows no alterations between WT and *Fam134b/c^dKO* mice either in both the number of NF200+ (G) and ChAT+ axons (H) or in their diameter distribution (I, J). $n = 3$ animals/group. Data information: Statistical significance was determined by one-way ANOVA (B–D) followed by Tukey's multiple comparisons test or Student's *t*-test (F–H). Data represent mean ± SEM. ns $P > 0.05$, *$P < 0.05$, **$P < 0.01$. Statistical analysis and exact p-values are included in the source data files. Source data are available online for this figure.

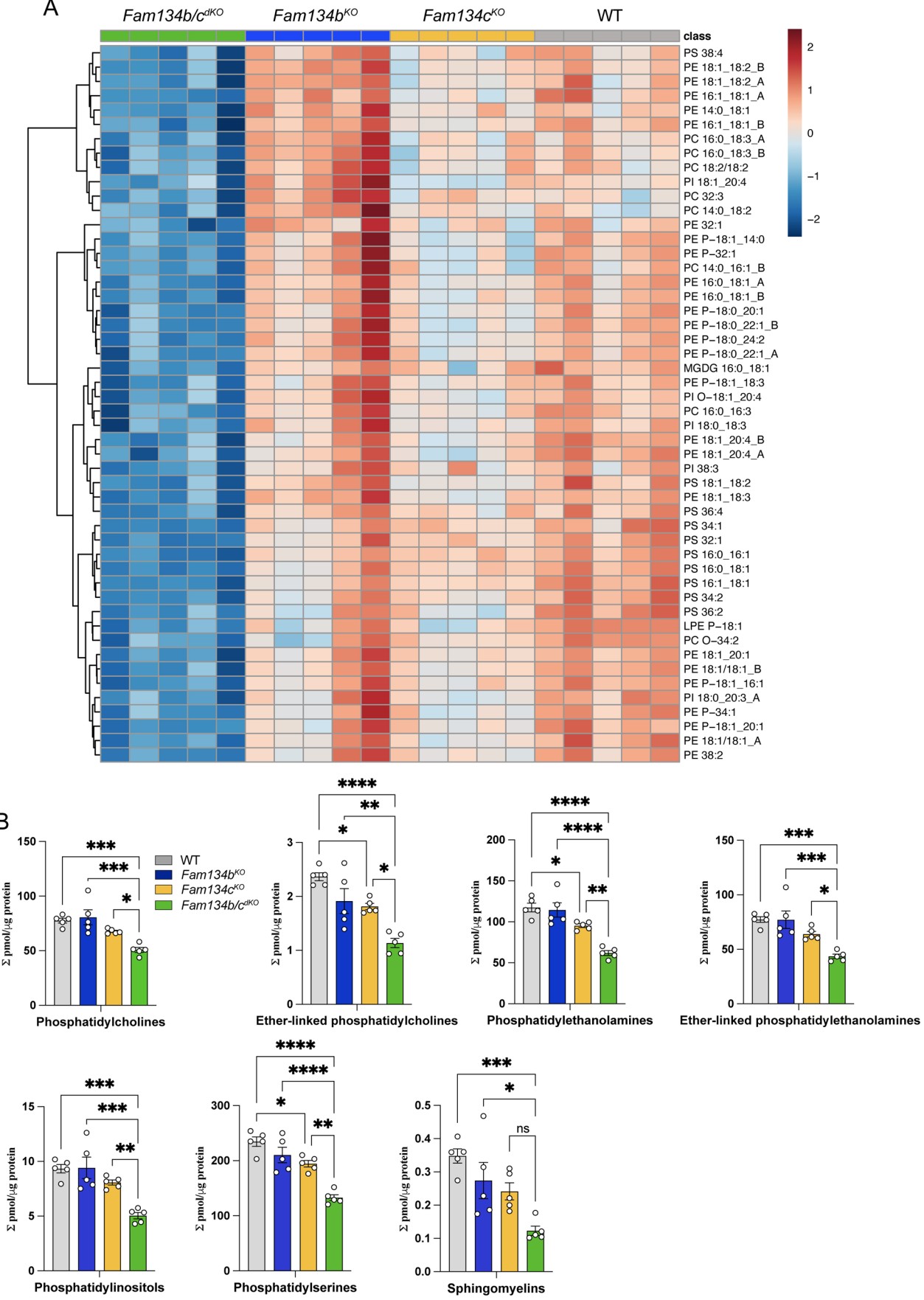

◄ **Figure EV5. Altered lipidome profile in *Fam134b/c^dKO* sciatic nerve.**

(A) Heatmap reporting the top 50 statistically significant lipids and comparing WT, *Fam134b^KO*, *Fam134c^KO*, and *Fam134b/c^dKO*. The log2 relative lipid quantity scale is depicted on the top right. (B) Lipid concentration in WT, *Fam134b^KO*, *Fam134c^KO*, and *Fam134b/c^dKO* sciatic nerves grouped in lipid subclasses. Lipid quantity is expressed in pmol and normalized to µg of total protein. $n = 5$ animals/group, 2 replicates/animal. Statistical significance was determined by one-way ANOVA followed by Tukey's multiple comparisons test. Data represent mean ± SEM. ns $P > 0.05$, *$P < 0.05$, **$P < 0.01$, ***$P < 0.001$, ****$P < 0.0001$. Statistical analysis and exact $P$ values are included in the source data files. Source data are available online for this figure.

