## [Peer Review File · EMBO Reports]

Fam134c and Fam134b shape axonal endoplasmic reticulum architecture in vivo.

Francescopaolo Iavarone, Marta Zaninello, Michela Perrone, Mariagrazia Monaco, Esther Barth, Felix Gaedke, Mariateresa Pizzo, Giorgia Di Lorenzo, Vincenzo Desiderio, Eduardo Sommella, Fabrizio Merciai, Emanuela Salviati, Pietro Campiglia, Livio Luongo, Elvira De Leonibus, Elena Rugarli, and Carmine Settembre

Corresponding author(s): Carmine Settembre (settembre@tigem.it), Elena Rugarli (elena.rugarli@uni-koeln.de), Francescopaolo Iavarone (f.iavarone@tigem.it), Elvira De Leonibus (elvira@deleonibus.it)

Review Timeline:

Submission Date:	9th Feb 24
Editorial Decision:	4th Mar 24
Revision Received:	28th May 24
Editorial Decision:	24th Jun 24
Revision Received:	27th Jun 24
Accepted:	3rd Jul 24

Transaction Report:

Dear Carmine,

Thank you for the submission of your research manuscript to our journal. We have now received the full set of referee reports that is copied below.

As you will see, the referees acknowledge that the findings are interesting and that the conclusions are overall supported by the data presented but they also raise a number of concerns and have suggestions how to further strengthen the data. The concerns relate to the potential different or overlapping functions of FAM134A/B/C, whether the different phenotypes observed might be caused by different expression levels, whether there is phylogenetic support for FAM134B and FAM134C as functional paralogs relative to FAM134A and why the phenotype you observe for single FAM134B KO mice differs from that reported before. The referees also ask for further analysis of the KO phenotypes, such as a more detailed analysis of ER morphology as well as alterations to mitochondrial morphology as well as autophagy. I feel that all these suggestions will further strengthen your work and the proposed experiments seem feasible and should be performed. While I agree with referee #1 that the analysis of tissue-specific functions of the FAM134 paralogs would be interesting, I also feel that this is beyond the scope of the current study and is thus not required for publication here. The mechanism by which FAM134A/B/C cause distinct phenotypes can be discussed in the manuscript. Further mechanistic insight (referee #1) is not required for publication in EMBO Reports.

Given these constructive comments, we would like to invite you to revise your manuscript with the understanding that the referee concerns (as detailed above and in their reports) must be fully addressed and their suggestions taken on board. Please address all referee concerns in a complete point-by-point response. Acceptance of the manuscript will depend on a positive outcome of a second round of review. It is EMBO Reports policy to allow a single round of revision only and acceptance or rejection of the manuscript will therefore depend on the completeness of your responses included in the next, final version of the manuscript.

We realize that it is difficult to revise to a specific deadline. In the interest of protecting the conceptual advance provided by the work, we recommend a revision within 3 months (June 4). Please discuss the revision progress ahead of this time with the editor if you require more time to complete the revisions.

I am also happy to discuss the revision further via e-mail or a video call, if you wish.

You can either publish the study as a short report or as a full article. With its current format, it will be published in our Scientific Reports section. For short reports, the revised manuscript should not exceed 27,000 characters (including spaces but excluding materials & methods and references) and 5 main plus 5 expanded view figures. The results and discussion sections must further be combined, which will help to shorten the manuscript text by eliminating some redundancy that is inevitable when discussing the same experiments twice. For a normal article there are no length limitations, but it should have more than 5 main figures and the results and discussion sections must be separate. In both cases, the entire materials and methods must be included in the main manuscript file.

*******IMPORTANT NOTE:**

We perform an initial quality control of all revised manuscripts before re-review. Your manuscript will FAIL this control and the handling will be delayed IN CASE the following APPLIES:

- 1) A data availability section providing access to data deposited in public databases is missing. If you have not deposited any data, please add a sentence to the data availability section that explains that.
- 2) Your manuscript contains statistics and error bars based on $n=2$. Please use scatter blots in these cases. No statistics should be calculated if $n=2$.

When submitting your revised manuscript, please carefully review the instructions that follow below. Failure to include requested items will delay the evaluation of your revision.*****

- 1) a .docx formatted version of the manuscript text (including legends for main figures, EV figures and tables). Please make sure that the changes are highlighted to be clearly visible.
- 2) individual production quality figure files as .eps, .tif, .jpg (one file per figure). Please download our Figure Preparation Guidelines (figure preparation pdf) from our Author Guidelines pages <https://www.embopress.org/page/journal/14693178/authorguide> for more info on how to prepare your figures.

4) a complete author checklist, which you can download from our author guidelines (<<https://www.embopress.org/page/journal/14693178/authorguide>>). Please insert information in the checklist that is also reflected in the manuscript. The completed author checklist will also be part of the RPF.

5) Please note that all corresponding authors are required to supply an ORCID ID for their name upon submission of a revised manuscript (<<https://orcid.org/>>). Please find instructions on how to link your ORCID ID to your account in our manuscript tracking system in our Author guidelines (<<https://www.embopress.org/page/journal/14693178/authorguide#authorshipguidelines>>)

6) We replaced Supplementary Information with Expanded View (EV) Figures and Tables that are collapsible/expandable online. A maximum of 5 EV Figures can be typeset. EV Figures should be cited as 'Figure EV1, Figure EV2' etc... in the text and their respective legends should be included in the main text after the legends of regular figures.

7) Before submitting your revision, primary datasets (and computer code, where appropriate) produced in this study need to be deposited in an appropriate public database (see <<https://www.embopress.org/page/journal/14693178/authorguide#dataavailability>>).

Specifically, we would kindly ask you to provide public access to the proteomics dataset.

The accession numbers and database should be listed in a formal "Data Availability " section (placed after Materials & Method) that follows the model below (see also <<https://www.embopress.org/page/journal/14693178/authorguide#dataavailability>>). Please note that the Data Availability Section is restricted to new primary data that are part of this study.

Data availability

Additional information on source data and instruction on how to label the files are available <<https://www.embopress.org/page/journal/14693178/authorguide#sourcedata>>.

10) Figure legends and data quantification:
The following points must be specified in each figure legend:

- the name of the statistical test used to generate error bars and P values,
- the number (n) of independent experiments (please specify technical or biological replicates) underlying each data point,
- the nature of the bars and error bars (s.d., s.e.m.)

- If the data are obtained from $n \leq 5$, show the individual data points in addition to the SD or SEM.
- If the data are obtained from $n \leq 2$, use scatter blots showing the individual data points.

Discussion of statistical methodology can be reported in the materials and methods section, but figure legends should contain a basic description of n , P and the test applied.

11) Our journal encourages inclusion of *data citations in the reference list* to directly cite datasets that were re-used and obtained from public databases. Data citations in the article text are distinct from normal bibliographical citations and should directly link to the database records from which the data can be accessed. In the main text, data citations are formatted as follows: "Data ref: Smith et al, 2001" or "Data ref: NCBI Sequence Read Archive PRJNA342805, 2017". In the Reference list, data citations must be labeled with "[DATASET]". A data reference must provide the database name, accession number/identifiers and a resolvable link to the landing page from which the data can be accessed at the end of the reference. Further instructions are available at [<https://www.embopress.org/page/journal/14693178/authorguide#referencesformat>](https://www.embopress.org/page/journal/14693178/authorguide#referencesformat).

12) All Materials and Methods need to be described in the main text. We would encourage you to use 'Structured Methods', our new Materials and Methods format. According to this format, the Materials and Methods section should include a Reagents and Tools Table (listing key reagents, experimental models, software and relevant equipment and including their sources and relevant identifiers) followed by a Methods and Protocols section in which we encourage the authors to describe their methods using a step-by-step protocol format with bullet points, to facilitate the adoption of the methodologies across labs. More information on how to adhere to this format as well as downloadable templates (.doc or .xls) for the Reagents and Tools Table can be found in our author guidelines: [<https://www.embopress.org/page/journal/14693178/authorguide#manuscriptpreparation>](https://www.embopress.org/page/journal/14693178/authorguide#manuscriptpreparation).

An example of a Method paper with Structured Methods can be found here: [<https://www.embopress.org/doi/10.15252/msb.20178071>](https://www.embopress.org/doi/10.15252/msb.20178071).

13) As part of the EMBO publication's Transparent Editorial Process, EMBO Reports publishes online a Review Process File to accompany accepted manuscripts. This File will be published in conjunction with your paper and will include the referee reports, your point-by-point response and all pertinent correspondence relating to the manuscript.

Kind regards,

Martina

Referee #1:

The endoplasmic reticulum (ER) homeostasis is crucial for cellular viability. Previous studies have demonstrated the significant role of ER autophagy in maintaining ER function and homeostasis. Among them, FAM134A, B, and C as receptors involved in ER autophagy have gained increasing attention in the field. While the loss of function of FAM134B has been shown to be

associated with HSAN-II disease, pathological investigations regarding other members of the FAM134 family are much less explored. In this manuscript, the authors attempt to uncover the functional correlations among the FAM134 family members and their importance in maintaining neuronal health.

This manuscript is abundant in data, with reliable results and exquisite formatting. It endeavors to tackle prominent issues in the field, rendering it well-suited for publication in the *Embo Reports* journal. At the same time, the draft also has flaws, such as: no explanation of the mechanism is given, some results are logically inconsistent. Additional experiments or direct responses to the reviewer's questions are needed.

Major comments

1 : In previous studies, the individual knockout of FAM134B alone can result in damage to sensory neurons and HSAN-II disease. However, in the author's manuscript, it appears that the simultaneous knockout of both FAM134B and FAM134C is required to observe this phenomenon. Is FAM134C associated with disease susceptibility? Are defects in FAM134C also present in patients with HSAN-II?

2 : In Figure 1, the author's behavioral experiments show differences appearing at 15 weeks of age, while analysis of neuronal electrical signals reveals significant differences as early as 4 weeks old. What is the reason for this?

3 : The author's series of observations on Fam134b/c dKO mice could be supplemented by tissue-specific gene knockouts, such as whether the degeneration of the sciatic nerve observed in Figure 2 and 3 is directly caused by damage to the neurons themselves.

4 : In Figure 3, the authors observe the degeneration of the triceps and gastrocnemius muscles in Fam134b/cdKO mice was due to neural degeneration or gene knockout in the muscles themselves? Are there signs of degeneration in other muscles? Are there defects in skeletal portions?

5: The authors should compare the changes in ER-phagy when FAM134B and FAM134C are individually knocked out versus knocked out simultaneously. Does the absence of either FAM134B or C result in the enhancement or weakening of the function of the other?

6: The authors observed a very distinct phenotype, which is exciting. But what is the underlying mechanism? Is this phenotype caused by the ER-phagy receptor function of the FAM134 family, or do the FAM134 proteins have other functions at different locations? If so, what does the author think they are?

Minor comments

1: Paragraph formatting is not uniform.

2: Why does simultaneous knockout of FAM134B and FAM134C lead to mouse mortality within 25 weeks?

3: Considering the significant role of the FAM134 family in ER-phagy, it is advisable to quantify the fragmentation of the endoplasmic reticulum and its surface area in neurons.

4: Given the accumulation effects observed in Fam134b/c dKO mice, can proteomic analysis be performed using Cre-induced knockout mice?

5: Can the authors supplement data from certain parts of the brain to determine if there are any defects?

Referee #2:

In this study, the authors investigate the FAM134 protein family, including FAM134A, FAM134B, and FAM134C, that mediates ER-phagy. While FAM134B mutations are linked to a form of hereditary sensory and autonomic neuropathy in humans, the physiological roles of the other FAM134 proteins remain unclear. The authors address this using single and combined knockout (KOs) in mice. Single KO in young mice show no major phenotypes, but combined Fam134b and Fam134c deletion (Fam134b/c dKO) leads to rapid neuromuscular and somatosensory degeneration, resulting in early death. Fam134b/c dKO triggered motor and sensory axon loss in the peripheral nervous system. Notably, Fam134b/c dKO mice exhibited disorganized tubular ER within long axons, without abnormalities in cortical ER. Cytoskeletal disorganization and impaired organelle distribution along Fam134b/c dKO axons indicated a failure in the structural role of ER in maintaining axonal organization. The authors conclude that FAM134C and FAM134B have critical functions in regulating motor and sensory axons by controlling tubular ER integrity and function.

Overall, the mouse work appears well done and described. The reorganization of ER in axons is quite robust (notably, similar morphologies have been seen in other recent studies). No specific mechanism for this reorganization is demonstrated, though several possibilities are discussed. I have a few comments regarding the interpretation of the results, and how they relate to recently published work.

1. Is there a phylogenetic support for Fam134b and Fam134c as functional paralogs relative to Fam134a? For instance, are they more similar to one another than to Fam134a, or do any species have single ortholog for Fam134b/c? Related to the above, it appears that the strong phenotype of the Fam134b/c dKO mice reflects protein amounts, i.e., is Fam134a simply less abundant, so its depletion is not as consequential. In fact, the authors describe this in sciatic nerve. The point I am trying to make is that all three FAM134 proteins might have similar functions. The authors' discussion seems to support this, and they do discuss in the conclusion the importance of further studies for Fam134a.

2. The periodic transverse ER ladders in the Zhu et al paper (HMG 2022) for At11K1/Reep1 null double mutant axons were striking in 3D reconstructions. Could the authors do the same to see if the appearance is similar in the Fam134b/c dKO axons?

It would be quite interesting if they are, given the different functions of the ER proteins in the two studies. The authors emphasize that the tubular ER is disorganized in axons, but in the Zhu et al. study although the ER was dramatically different, it wasn't disorganized; it has a new type of organization. Is the same true here? That study also noted fragmentation of mitochondria. Could that be assessed here?

3. The Zhu et al HMG paper reported specific alterations in phosphorylation of neurofilaments. Could the authors assess this also, and perhaps levels of other cytoskeletal proteins and modifications of those in a similar fashion (the authors address this issue to some degree, in a different way, in their proteomics study)? Again, the comparison could be quite interesting given the apparently similar reorganization of ER.

Referee #3:

lavarone et al in this study reported on the single and combined loss of the ERphagy receptors FAM134A-C in constitutive knockout mice. They convincingly demonstrate that combined loss of FAM134B and FAM134C severely impact viability, somatosensory defects, and severe motoneuron degeneration, e.g. motor and sensory axon loss in the PNS. These phenotypes correlate at the ultrastructural level with tubular membrane accumulations that most likely represent tubular ER accumulations. Finally, proteomic analyses show that proteins related to ER organization as well as some cytoskeletal are altered in sciatic nerves from DKO mice.

Overall, this is an important study revealing for the first time the physiological consequences of organismal loss of a major class of ERphagy receptors. The study is well executed and written and the data are largely compelling. I thus only have a few points to be addressed.

1. Alterations in axonal ER: From the example images provided in Fig EV4 is unclear to me how the authors identified the accumulated membranes as ER for the data shown in Fig. 4B,C. In light of the proteomic data, it appears that while some ER proteins accumulate, major tubular ER proteins such as REEP1/2 are lost or reduced in sciatic nerve preparations. It hence would be important to verify that the accumulated membranes shown in Figs 4+EV4 indeed correspond to ER. While this may be technically challenging to demonstrate at the ultrastructural level I would at least want to see accumulation of tubular ER markers in axons at the confocal microscopy level of analysis.

2. Alterations in sciatic nerve proteome: The proteomics dataset in Fig 5 is interesting but in my view would require some level of verification, for example at the level of immunoblotting of sciatic nerves as illustrated in panel A of the same figure to confirm the key messages. I also wonder whether any alterations in the levels or localization of other ERphagy receptors or general components of the autophagy pathway were observed in DKO samples.

3. Electrophysiology: I am not an expert in nociception but remain puzzled by the fact that sensation of heat pain is reduced in DKOs, while recordings from nociceptive neurons of the spinal cord dorsal horn suggest hyperexcitability. These data thus seem to clash. It is also not clear from the description which type of fibers were recorded. A better description of these experiments and clarification as to how the data obtained relate to the behavioral phenotype would be important. Do the authors have evidence that the increased firing rate and duration of excitation is related to altered calcium levels?

Minor:

Figure 2: Higher magnified views to illustrate the axonal degeneration phenotype would aid the message.

Referee #1:

The endoplasmic reticulum (ER) homeostasis is crucial for cellular viability. Previous studies have demonstrated the significant role of ER autophagy in maintaining ER function and homeostasis. Among them, FAM134A, B, and C as receptors involved in ER autophagy have gained increasing attention in the field. While the loss of function of FAM134B has been shown to be associated with HSAN-II disease, pathological investigations regarding other members of the FAM134 family are much less explored. In this manuscript, the authors attempt to uncover the functional correlations among the FAM134 family members and their importance in maintaining neuronal health.

This manuscript is abundant in data, with reliable results and exquisite formatting. It endeavors to tackle prominent issues in the field, rendering it well-suited for publication in the Embo Reports journal. At the same time, the draft also has flaws, such as: no explanation of the mechanism is given, some results are logically inconsistent. Additional experiments or direct responses to the reviewer's questions are needed.

Major comments

1 : In previous studies, the individual knockout of FAM134B alone can result in damage to sensory neurons and HSAN-II disease. However, in the author's manuscript, it appears that the simultaneous knockout of both FAM134B and FAM134C is required to observe this phenomenon. Is FAM134C associated with disease susceptibility? Are defects in FAM134C also present in patients with HSAN-II?

The Fam134b knockout (Fam134bKO) mouse, characterized in previous studies, exhibits a late onset of sensory disease. We analyzed single and combined knockouts (KOs) at earlier time points and found that only *Fam134b/c^{dKO}* mice show an early-onset disease affecting both sensory and motor axons. We concluded that Fam134c partially compensates for the loss of Fam134b in sensory neurons and completely compensates in motor neurons in HSAN-II. To our knowledge, Fam134c

mutations have not yet been associated with susceptibility to motor or sensory diseases in humans. We have discussed these considerations in the manuscript as follows: “Given that loss-of-function mutations in FAM134B exclusively affect the sensory component of the peripheral nervous system in humans and mice (Khaminets *et al*, 2015; Kurth *et al*, 2009), our work suggests the potential role of FAM134C in preserving motor functionality in patients with FAM134B mutations. Furthermore, considering the *Fam134b*^{KO} mouse model shows a late onset of the sensory disease (Khaminets *et al.*, 2015), the early onset and fast progression of *Fam134b/c*^{dKO} phenotype led us to infer that Fam134c may prevent the anticipation and exacerbation of the HSN axonopathy in mouse. This observation also prompts the intriguing possibility of targeting FAM134C activation as a potential therapeutic strategy for HSNII”.

2 : In Figure 1, the author's behavioral experiments show differences appearing at 15 weeks of age, while analysis of neuronal electrical signals reveals significant differences as early as 4 weeks old. What is the reason for this?

Most of the behavioral experiments show defects in *Fam134b/c*^{dKO} mice as early as 4 weeks, with a significant worsening by 15 weeks (Fig. 1H, J-L and EV1H). The open field test, which evaluates exploratory behavior and general mobility, showed a significant reduction only at 15 weeks (Fig. EV1E-G). We also observed that most axons in the *Fam134b/c*^{dKO} nerve had undergone massive degeneration by 15 weeks; conversely, only initial signs of degeneration were evident at 4 weeks (Fig. 2A-D). For these reasons, it seems reasonable to perform in vivo electrophysiology tests on nociceptive neurons in 4-week-old mice.

3 : The author's series of observations on *Fam134b/c* dKO mice could be supplemented by tissue-specific gene knockouts, such as whether the degeneration of the sciatic nerve observed in Figure 2 and 3 is directly caused by damage to the neurons themselves.

We agree with the reviewer that future studies are necessary to dissect the contribution of each Fam134 protein in specific functions and tissues of the body. However, the primary aim of our work is to provide the first in vivo analysis of the Fam134 protein family function. Therefore, we believe that the reviewer's request is beyond the scope of this manuscript. The editor has concurred with this assessment.

4 : In Figure 3, the authors observe the degeneration of the triceps and gastrocnemius muscles in Fam134b/cdKO mice was due to neural degeneration or gene knockout in the muscles themselves? Are there signs of degeneration in other muscles? Are there defects in skeletal portions?

We have extended the phenotypic analysis to additional skeletal muscles (tibialis anterior and quadriceps) (not shown), cardiac muscle (Fig. 1D) and skeletal elements (femur, tibia, humerus, and scapula) (Fig. 1E). We found a significant reduction in the weight and size, respectively, of all analyzed *Fam134b/c^{dKO}* muscles and bones compared with WT (Fig. 1D, E).

We agree with the reviewers that we cannot formally demonstrate that the muscle phenotype is due exclusively to neuromuscular junction degeneration, thus, in the revised version, we have reorganized the text and figures, combining all the skeletal and muscle phenotypic data in Fig. 1.

5: The authors should compare the changes in ER-phagy when FAM134B and FAM134C are individually knocked out versus knocked out simultaneously. Does the absence of either FAM134B or FAM134C result in the enhancement or weakening of the function of the other?

The analysis of the ER membrane marker Calnexin (an ER-phagy substrate) by Western blot revealed a significant increase in Calnexin levels in *Fam134b/c^{dKO}* compared with WT, *Fam134b^{KO}*, and *Fam134c^{KO}* sciatic nerves (Fig. EV7A, C). This observation indicates that ER-phagy impairment in the *Fam134b/c^{dKO}* is more severe than in the single KO. Consistently, the accumulation of ER membrane in

Fam134b/c^{dKO} was more severe than in *Fam134b^{KO}* and *Fam134c^{KO}* sciatic nerves (Fig. 4A-B).

These data clearly demonstrate that FAM134B and FAM134C can partly compensate for each other during ER-phagy. These *in vivo* observations confirm recent *in vitro* studies from the Harper's laboratory (Hoyer et al., 2023).

6: The authors observed a very distinct phenotype, which is exciting. But what is the underlying mechanism? Is this phenotype caused by the ER-phagy receptor function of the FAM134 family, or do the FAM134 proteins have other functions at different locations? If so, what does the author think they are?

Given the similarities between mouse models carrying *Fam134b/c* or *Alt1^{K1/K1}/Reep1^{-/-}* mutations (Zhu et al, 2022), and considering that the latter two proteins are not ER-phagy receptors, we discussed in the text that *Fam134b* and *Fam134c* might also have ER-phagy-independent roles. However, investigating these possible mechanisms *in vivo* is very difficult, and future *in vitro* studies will be instrumental in addressing these questions. Thus, in agreement with the editor, we will not address this point in the current manuscript.

Minor comments

1: Paragraph formatting is not uniform.

We have uniformed paragraph formatting.

2: Why does simultaneous knockout of FAM134B and FAM134C lead to mouse mortality within 25 weeks?

We believe that lethality associated to *Fam134b/c^{dKO}* phenotype is very likely due to the progressive motor and sensory degeneration.

3: Considering the significant role of the FAM134 family in ER-phagy, it is advisable to quantify the fragmentation of the endoplasmic reticulum and its surface area in neurons.

Our data in Fig. 4 indicate an aberrant ER accumulation as a consequence of *Fam134b/c*^{dKO}. We extended these results by generating a 3D ER reconstruction in longitudinal axons, demonstrating that not only the ER accumulates, but it also redistributes, forming “ladder-like” structures in the *Fam134b/c*^{dKO} that are easily appreciable compared to WT (Fig. 4). We thus concluded that the lack of Fam134b and Fam134c, more so than in single KOs, results in ER reorganization, completely changing both its surface area and structure.

4: Given the accumulation effects observed in *Fam134b/c* dKO mice, can proteomic analysis be performed using Cre-induced knockout mice?

We understand the reviewer's point and agree that it would be interesting to address. However, generating and analyzing new mouse models with inducible Cre system would require at least two years of experiments. We are definitely interested in investigating this in our future experiments. Thus, in agreement with the editor, this concern has not been addressed in the current manuscript.

5: Can the authors supplement data from certain parts of the brain to determine if there are any defects?

Prompted by reviewer's request we analyzed sagittal brain sections from young WT and *Fam134b/c*^{dKO} mice and found no major alterations in tissue architecture (Appendix Fig. S2). Immunofluorescence analysis for astrogliosis, microglia activation, and neurons using GFAP, Iba1, and NeuN markers, respectively, highlighted no significant changes in terms of neuroinflammation or neuronal density (Appendix Fig. S3A-B). Additionally, the dendrite marker MAP2 was similarly distributed in the granular cell layer of the WT and *Fam134b/c*^{dKO} cerebellum (Appendix Fig. S3C), excluding the possibility of decreased neuroplasticity and structural integrity in the cerebellar cortex. We also found no notable differences in motoneuron density in *Fam134b/c*^{dKO} mice at 4 weeks of age (Appendix Fig. S4A, B). Similarly, the density of different types of sensory neurons in the DRG of *Fam134b/c*^{dKO} mice was comparable to that in WT mice (Appendix Fig. S4C, D).

Collectively, these observations strongly suggest that the primary phenotype is due to the degeneration of the peripheral nervous system.

Referee #2:

In this study, the authors investigate the FAM134 protein family, including FAM134A, FAM134B, and FAM134C, that mediates ER-phagy. While FAM134B mutations are linked to a form of hereditary sensory and autonomic neuropathy in humans, the physiological roles of the other FAM134 proteins remain unclear. The authors address this using single and combined knockout (KOs) in mice. Single KO in young mice show no major phenotypes, but combined Fam134b and Fam134c deletion (Fam134b/c dKO) leads to rapid neuromuscular and somatosensory degeneration, resulting in early death. Fam134b/c dKO triggered motor and sensory axon loss in the peripheral nervous system. Notably, Fam134b/c dKO mice exhibited disorganized tubular ER within long axons, without abnormalities in cortical ER. Cytoskeletal disorganization and impaired organelle distribution along Fam134b/c dKO axons indicated a failure in the structural role of ER in maintaining axonal organization. The authors conclude that FAM134C and FAM134B have critical functions in regulating motor and sensory axons by controlling tubular ER integrity and function.

Overall, the mouse work appears well done and described. The reorganization of ER in axons is quite robust (notably, similar morphologies have been seen in other recent studies). No specific mechanism for this reorganization is demonstrated, though several possibilities are discussed. I have a few comments regarding the interpretation of the results, and how they relate to recently published work.

1. Is there a phylogenetic support for Fam134b and Fam134c as functional paralogs relative to Fam134a? For instance, are they more similar to one another than to Fam134a, or do any species have single ortholog for Fam134b/c? Related to the above, it appears that the strong phenotype of the Fam134b/c dKO mice reflects protein amounts, i.e., is Fam134a simply less abundant, so its depletion is not as consequential. In fact, the authors describe this in sciatic nerve. The point I am trying

to make is that all three FAM134 proteins might have similar functions. The authors' discussion seems to support this, and they do discuss in the conclusion the importance of further studies for Fam134a.

The three Fam134 members have similar functions, although they exhibit different kinetics and specific regulations (Di Lorenzo *et al*, 2022; Gonzalez *et al*, 2023; Reggio *et al*, 2021). We did not find phylogenetic evidence indicating that Fam134b is more similar to Fam134c than to Fam134a. A recent in vitro study corroborate our hypothesis that the copy number of FAM134 proteins, rather than the functional specificity of each Fam134 isoform, is crucial for maintaining proper ER homeostasis in axons of cultured neurons (Hoyer *et al*, 2023). In agreement with these conclusions, we believe that the severe phenotype of the *Fam134b/c^{dKO}* mice reflects the very low abundance of Fam134a in the sciatic nerve.

2. The periodic transverse ER ladders in the Zhu et al paper (HMG 2022) for *At11Kl/Reep1* null double mutant axons were striking in 3D reconstructions. Could the authors do the same to see if the appearance is similar in the *Fam134b/c* dKO axons? It would be quite interesting if they are, given the different functions of the ER proteins in the two studies. The authors emphasize that the tubular ER is disorganized in axons, but in the Zhu et al. study although the ER was dramatically different, it wasn't disorganized; it has a new type of organization. Is the same true here? That study also noted fragmentation of mitochondria. Could that be assessed here?

We imaged the ER structure in sagittal axons from WT and *Fam134b/c^{dKO}* sciatic nerve sections using electron microscopy. A 3D reconstruction was then performed following ER membrane segmentation. The results clearly indicated the presence of a "ladder-like" ER structure in *Fam134b/c^{dKO}* axons compared with WT (Fig. 4D, E), similar to the one observed by Zhu et al. (Zhu *et al*, 2022).

We did not analyze in detail the mitochondria morphology, as we did not observe an obvious phenotype from EM analysis.

We agree with the reviewer that it is very interesting how the lack of proteins with different predicted functions can lead to similar phenotypes. We discussed these data and the various possibilities in the discussion section.

3. The Zhu et al HMG paper reported specific alterations in phosphorylation of neurofilaments. Could the authors assess this also, and perhaps levels of other cytoskeletal proteins and modifications of those in a similar fashion (the authors address this issue to some degree, in a different way, in their proteomics study)? Again, the comparison could be quite interesting given the apparently similar reorganization of ER.

We analyzed both the total and hyperphosphorylated forms of the Neurofilament heavy chain (NF200) by Western blot in the sciatic nerve of 2-week-old mice (Fig. 5H, I). We observed a shifted NF200 band in *Fam134b/c^{dKO}* samples using the total antibody, suggesting decreased phosphorylation levels. Indeed, using the antibody for hyperphosphorylated NF200, we found a blunted signal in *Fam134b/c^{dKO}* samples compared with other genotypes (Fig. 5H, I), confirming a strong decrease in NF200 phosphorylation in the *Fam134b/c^{dKO}* sciatic nerve. Considering that in healthy mature axons, NF200 is highly phosphorylated to ensure proper neurofilament assembly (Laser-Azogui et al., 2015), these results clearly indicate the presence of dysfunctional NF200 in the *Fam134b/c^{dKO}* sciatic nerve.

Western blot analysis of other cytoskeletal proteins, such as beta-III-tubulin and beta-actin, showed no changes (Fig. 5H, I), suggesting that only specific cytoskeletal components were affected by the *Fam134b/c^{dKO}* phenotype. Notably, the finding of decreased NF200 levels in *Fam134b/c^{dKO}*, along with the reduced levels of Reep1 and Reep2, validated some of the most downregulated proteins in our proteomic data (Fig. 5F, G).

Referee #3:

lavarone et al in this study reported on the single and combined loss of the ERphagy receptors FAM134A-C in constitutive knockout mice. They convincingly demonstrate that combined loss of FAM134B and FAM134C severely impact viability,

somatosensory defects, and severe motoneuron degeneration, e.g. motor and sensory axon loss in the PNS. These phenotypes correlate at the ultrastructural level with tubular membrane accumulations that most likely represent tubular ER accumulations. Finally, proteomic analyses show that proteins related to ER organization as well as some cytoskeletal are altered in sciatic nerves from DKO mice.

Overall, this is an important study revealing for the first time the physiological consequences of organismal loss of a major class of ERphagy receptors. The study is well executed and written and the data are largely compelling. I thus only have a few points to be addressed.

1. Alterations in axonal ER: From the example images provided in Fig EV4 is unclear to me how the authors identified the accumulated membranes as ER for the data shown in Fig. 4B,C. In light of the proteomic data, it appears that while some ER proteins accumulate, major tubular ER proteins such as REEP1/2 are lost or reduced in sciatic nerve preparations. It hence would be important to verify that the accumulated membranes shown in Figs 4+EV4 indeed correspond to ER. While this may be technically challenging to demonstrate at the ultrastructural level I would at least want to see accumulation of tubular ER markers in axons at the confocal microscopy level of analysis.

We improved our electron microscopy analysis and performed a 3D reconstruction of the axonal ER in *Fam134b/c*^{dKO} and WT mice. The results clearly indicated a strong accumulation of ER membrane in *Fam134b/c*^{dKO} compared with WT (Fig. 4D, E). Notably, ER structure also is completely reorganized and assume a ladder-like shape in *Fam134b/c*^{dKO} phenotype (Fig. 4E). These results appeared very similar to the phenotype observed in the *Atf1*^{K1/K1}/*Reep1*^{-/-} mice showed in (Zhu et al., 2022).

2. Alterations in sciatic nerve proteome: The proteomics dataset in Fig 5 is interesting but in my view would require some level of verification, for example at the level of immunoblotting of sciatic nerves as illustrated in panel A of the same figure to confirm the key messages. I also wonder whether any alterations in the levels or

localization of other ERphagy receptors or general components of the autophagy pathway were observed in DKO samples.

We validated our proteomic analysis by Western blot, confirming the decrease of Neurofilament heavy chain (NF200), Reep1, and Reep2 in 2-week-old *Fam134b/c^{dKO}* sciatic nerve compared with other genotypes (Fig. 5F-I). We then analyzed the protein levels of other ER-phagy receptors (Alt3, Ccp1, Tex264, and Rtn3) and found no significant alterations, except in Rtn3 levels, which showed a decrease in *Fam134b/c^{dKO}* compared with other genotypes (Fig. EV7A, D). Rtn3, along with Reep proteins, localizes preferentially in tubular ER and participates in ER tubulation. Notably, the altered ER organization we found in *Fam134b/c^{dKO}* sciatic nerve (Fig. 4A-E) might be, at least in part, a reasonable consequence of the decrease in these proteins.

Furthermore, Western blot analysis of the autophagy markers Sqstm1/p62 and Lc3 revealed no remarkable differences (Fig. EV7A, B), indicating that general autophagy was not affected in any of the genotypes analyzed.

3. Electrophysiology: I am not an expert in nociception but remain puzzled by the fact that sensation of heat pain is reduced in DKO, while recordings from nociceptive neurons of the spinal cord dorsal horn suggest hyperexcitability. These data thus seem to clash. It is also not clear from the description which type of fibers were recorded. A better description of these experiments and clarification as to how the data obtained relate to the behavioral phenotype would be important. Do the authors have evidence that the increased firing rate and duration of excitation is related to altered calcium levels?

There are several peripheral neuropathies, such as HSAN and diabetic neuropathy, in which the perception of pain becomes abnormal and the discrimination between painful and non-painful stimuli is impaired. The hot plate test measures the thermal nociceptive response. On the other hand, peripheral neuropathies are also associated with abnormal tactile pain perception. For example, diabetic patients can have reduced thermal pain and increased tactile allodynia (Jolivald et al., 2020). In this study, we measured the response of dorsal horn spinal cord neurons after a

mechanical stimulus. Our results suggest that, similar to diabetic-related peripheral neuropathy, our mice have reduced thermal pain but increased tactile pain perception. We used in vivo electrophysiology in order to avoid possible influences due to motor impairment observed in this transgenic mouse phenotype.

Minor:

Figure 2: Higher magnified views to illustrate the axonal degeneration phenotype would aid the message.

We have provided a higher magnification for all the pictures in Figure 2.

References

Di Lorenzo G, Iavarone F, Maddaluno M, Plata-Gomez AB, Aureli S, Quezada Meza CP, Cinque L, Palma A, Reggio A, Cirillo C *et al* (2022) Phosphorylation of FAM134C by CK2 controls starvation-induced ER-phagy. *Sci Adv* 8: eabo1215

Gonzalez A, Covarrubias-Pinto A, Bhaskara RM, Glogger M, Kuncha SK, Xavier A, Seemann E, Misra M, Hoffmann ME, Brauning B *et al* (2023) Ubiquitination regulates ER-phagy and remodelling of endoplasmic reticulum. *Nature* 618: 394-401

Hoyer MJ, Capitanio C, Smith IR, Paoli JC, Bieber A, Jiang Y, Paulo JA, Gonzalez-Lozano MA, Baumeister W, Wilfling F *et al* (2023) Combinatorial selective ER-phagy remodels the ER during neurogenesis. *bioRxiv*

Khaminets A, Heinrich T, Mari M, Grumati P, Huebner AK, Akutsu M, Liebmann L, Stolz A, Nietzsche S, Koch N *et al* (2015) Regulation of endoplasmic reticulum turnover by selective autophagy. *Nature* 522: 354-358

Kurth I, Pamminger T, Hennings JC, Soehendra D, Huebner AK, Rotthier A, Baets J, Senderek J, Topaloglu H, Farrell SA *et al* (2009) Mutations in FAM134B, encoding a newly identified Golgi protein, cause severe sensory and autonomic neuropathy. *Nat Genet* 41: 1179-1181

Reggio A, Buonomo V, Berkane R, Bhaskara RM, Tellechea M, Peluso I, Polishchuk E, Di Lorenzo G, Cirillo C, Esposito M *et al* (2021) Role of FAM134 paralogues in endoplasmic reticulum remodeling, ER-phagy, and Collagen quality control. *EMBO Rep* 22: e52289

Vlachos I, Herry C, Lüthi A, Aertsen A, Kumar A (2011) Context-Dependent Encoding of Fear and Extinction Memories in a Large-Scale Network Model of the Basal Amygdala. *PLOS Computational Biology* 7: e1001104

Zhu PP, Hung HF, Batchenkova N, Nixon-Abell J, Henderson J, Zheng P, Renvoise B, Pang S, Xu CS, Saalfeld S *et al* (2022) Transverse endoplasmic reticulum expansion in hereditary spastic paraplegia corticospinal axons. *Hum Mol Genet* 31: 2779-2795

Dear Carmine,

Thank you for the submission of your revised manuscript to EMBO reports. We have now received the full set of referee reports that is copied below.

As you will see, referee #3 supports publication without further revisions. Referee #1 asked for further epistasis experiments, i.e., to express FAM134C in the FAM134B KO background to check for compensation. I fully agree that this would be an informative experiment but also feel that this is beyond the scope at this stage. Please discuss this point in the manuscript and in a point-by-point response.

Browsing through the manuscript myself, I noticed a few editorial things that we need before we can proceed with the official acceptance of your study.

1) Your manuscript will be published in our Reports section. For short reports, the revised manuscript should not exceed 27,000 characters (including spaces but excluding materials & methods and references) and 5 main plus 5 expanded view figures. The results and discussion sections must further be combined, which will help to shorten the manuscript text by eliminating some redundancy that is inevitable when discussing the same experiments twice.

2) Please provide up to 5 keywords.

3) Please provide the e-mail addresses of all corresponding authors on the title page.

4) Regarding the Author Contributions, we now use CRediT to specify the contributions of each author in the journal submission system. Therefore, please remove the Author Contributions from the manuscript file and make sure that the author contributions in our manuscript tracking system are correct and up-to-date. The information you specified in the system will be automatically retrieved and typeset into the article. You can enter additional information in the free text box provided, if you wish.

5) The ORCID ID for corresponding author Dr. De Leonibus is still missing. Please find instructions on how to link your ORCID ID to your account in our manuscript tracking system in our Author guidelines (<<https://www.embopress.org/page/journal/14693178/authorguide#authorshipguidelines>>)

6) Dataset EV1 and EV2: Please provide each legend as a separate tab/sheet in the respective Excel file instead of an additional .txt file. Then upload the .xls file as Dataset EV# w/o zipping.

7) Dataset EV3 is not a complex table but rather a "normal" table and should be included as Table 3 in the manuscript.

8) Movies: please provide individual movie files and individual legends, each movie zipped with its legend. Instead of referring to Movie EV1 in the text, you can then refer to Movie EV1-EV8, e.g. Thank you.

9) The callout to "Suppl. Fig. 1a" in Table 1 needs to be updated to the correct callout.

10) Appendix: I noticed a typo in the legend of Appendix Figure S3 with "immune uorescence" instead of "immunofluorescence". And missing comma in the legend of Appendix Figure S5 (...showing degenerating axons, accumulating organelles, and cytoskeletal ...).

11) Please note that we can only typeset up to 5 EV figures. Please move some of the figures to the Appendix.

12) Author Checklist:

- "Materials - Newly created materials" only refers to reagents or e.g. KO mice you generated in this particular paper.
- If you change Dataset EV3 to Table 3, please do not forget to update the reference in the Author Checklist.

13) Animal experiments: please provide the reference number for the approval (in addition to the authority granting ethics approval) in the Methods section.

14) Data availability section:

- Please provide a link that directly resolves to the Dataset PXD050983 on PRIDE.
- Just a reminder not to forget to remove the reviewer password.

15) Materials and Methods should be Methods.

16) We perform a routine image check on all manuscripts prior to publication. In this context we detected the following aberrations. Please carefully check the composition of these figures.

- Figure 1G and EV1D: the mice shown in Fig. 1G seem to be shown again in Fig. EV1D (WT in 1G seems to correspond to WT and Fam134b KO in EV1D. Fam134b/c dKO in 1G seems to correspond to Fam134b/c dKO in EV1D).

- Figure 2I, J and Appendix Figure S5A, B: it seems that the same WT controls are shown in both figures and the Fam134b/c dKO images shown in 2I/J are repeated in S5. In case the duplication is intentional, please state so in the respective Figure legends. If you wish to show representative images and variation/reproducibility, I strongly suggest choosing different images to showcase the full spectrum.

- Figure 4A and Figure EV4A: the EM image for Fam134b/c dKO in Fig. 4A is shown again in Fig. EV4A (accumulated, bottom image). Please check and if you wish to show variation, please include another example image.

17) Source data

- Source data for Fig. 1F, G are missing.

- Please upload the Source Data as one folder per figure, instead of the zipped file containing all figures.

18) Hoyer et al 2023 is a preprint. Please cite it in the text as (preprint: Hoyer et al, 2023) and add [PREPRINT] at the end of the reference in the reference list. We also need the DOI of the paper on bioRxiv.

19) Our production/data editors have asked you to clarify several points in the figure legends (see below). Please incorporate these changes in the manuscript and return the revised file with tracked changes with your final manuscript submission.

- Please note that the figure 2i-j are mislabeled as figure 2d-e in the manuscript. This needs to be rectified.

- Please note that the figure 3a is mislabeled as figure 3b, for statistical test information, in the manuscript. This needs to be rectified.

- Please note that the figure EV 3a is mislabeled as figure EV 3b, for statistical test information, in the manuscript. This needs to be rectified.

- Please note that the error bars are not defined in the legends of figures EV 5a-b.

- Please note that we require the specification of the exact p-values instead of the range (e.g. $p=0.0348$ instead of $p < 0.05$). Please carefully check the following figure panels and provide the the exact p values: figures 1a, c-e, h, j-l; 2b-d, f-h; 3a, c-f, h-k; 4b-c; 5g, i; EV 1a, c, e-f, h-k; EV 2c; EV 3a, e-f, h-k; EV 7c; EV 8b.

- If you define criteria that are either valid for several panels in the figure or that provide a list of definitions for different panels in that figure (e.g., "Statistical significance was determined by One-way ANOVA ($p=0.0026$ (A), $p=0.1975$ (C) etc.") then add a prefix "Data Information: " so that it is clear that the description that follows does not belong to the last panel you described.

20) Abstract: please describe your findings in present tense. It might be worth mentioning dKOs with Fam134a in the abstract and the absence of a phenotype for these.

21) Finally, EMBO Reports papers are accompanied online by

A) a short (1-2 sentences) summary of the findings and their significance,

B) 2-3 bullet points highlighting key results and

C) a schematic summary figure that provides a sketch of the major findings (not a data image).

Please provide the summary figure as a separate file in PNG or JPG format at a size of 550x300-600 pixels (width x height).

Please note that the size is rather small and that text needs to be readable at the final size. Please send us this information along with the revised manuscript.

22) On a different note, I would like to alert you that EMBO Press offers a new format for a video-synopsis of work published with us, which essentially is a short, author-generated film explaining the core findings in hand drawings, and, as we believe, can be very useful to increase visibility of the work. This has proven to offer a nice opportunity for exposure i.p. for the first author(s) of the study. Please see the following link for representative examples and their integration into the article web page:

<https://www.embopress.org/doi/full/10.15252/emj.2019103932>

With kind regards,

Martina

Referee #1:

The author has addressed the reviewers' questions, resulting in a significantly improved manuscript. I believe that after clarifying the following points, the manuscript will be suitable for publication in EMBO Reports.

- 1: In all the experiments conducted by the author, mice with a single knockout of FAM134B appear healthy.
- 2: To verify whether FAM134C can partially compensate for the function of FAM134B under physiological and pathological conditions, the author should overexpress FAM134C in FAM134B knockout mice to determine if there is any alleviation.

Referee #3:

The authors have addressed all my previous comments and questions. Thus I recommend publication of this very nice piece of work.

Referee #1:

The author has addressed the reviewers' questions, resulting in a significantly improved manuscript. I believe that after clarifying the following points, the manuscript will be suitable for publication in EMBO Reports.

1: In all the experiments conducted by the author, mice with a single knockout of FAM134B appear healthy.

2: To verify whether FAM134C can partially compensate for the function of FAM134B under physiological and pathological conditions, the author should overexpress FAM134C in FAM134B knockout mice to determine if there is any alleviation.

We agree with the reviewer that further studies *in vivo* would reinforce the concept of FAM134C compensation in absence of FAM134B, in both physiological and pathological conditions. However, the primary aim of our work is to provide the first *in vivo* analysis of the Fam134 protein family function. Therefore, we believe that the reviewer's request is beyond the scope of this manuscript. The editor has concurred with discussing this concern in the manuscript. We discussed this point in the manuscript as follow: "The early onset and rapid progression of the *Fam134b/c^{dKO}* phenotype suggest that Fam134c may delay or mitigate the severity of HSAN2 axonopathy in mice. This observation opens the possibility of targeting FAM134C activation as a therapeutic strategy for HSAN2."

Dr. Carmine Settembre
Telethon Institute of Genetics and Medicine (TIGEM)
Federico II University, Naples
Department of Translational Medicine
Via campi flegrei 34
Pozzuoli 80070
Italy

Dear Carmine,

Thank you for incorporating the final minor changes. I am very pleased to accept your manuscript for publication in the next available issue of EMBO reports. Thank you for your contribution to our journal.

Best regards,

Martina
